# De Sitter scattering amplitudes in the Born approximation

**Renata Ferrero**[*] and **Chris Ripken**[†]

Institute of Physics (THEP), Johannes Gutenberg-Universität,
Staudingerweg 7, 55128 Mainz, Germany

[*] rferrero@uni-mainz.de, [†] aripken@uni-mainz.de

## Abstract

We present a covariant framework to compute scattering amplitudes and potentials in a de Sitter background. In this setting, we compute the potential of a graviton-mediated scattering process involving two very massive scalars at tree level. Although the obtained scattering potential reproduces the Newtonian potential at short distances, on Hubble-size length scales it is affected by the constant curvature: effectively, it yields a repulsive force at sub-Hubble distances. This can be attributed to the expansion of the de Sitter universe. Beyond the de Sitter horizon, the potential vanishes identically. Hence, the scattering amplitude unveils the geometric properties of de Sitter spacetime in a novel and nontrivial way.

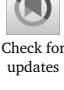
# 1  Introduction

Cosmological observations suggest that we are living in a universe with an accelerating expansion. Evidence for this is provided by observations of far-away galaxies [1, 2] and of the Cosmic Microwave Background [3]. In General Relativity (GR), this is modeled by a de Sitter (dS) universe: the unique constantly curved solution to Einstein's equation with positive cosmological constant.

Reconciling GR with quantum theory is one of the main challenges in modern theoretical physics. Perturbative quantisation in a flat background leads to a theory that is perturbatively nonrenormalisable [4–6], which fails to be predictive. Many proposals to cure this, and define a theory of Quantum Gravity have been put forward. Prominent examples are string theory [7], loop quantum gravity [8, 9], and asymptotically safe quantum gravity, using continuum [10–15] and discrete [16, 17] methods, respectively. However, so far no approach has been unequivocally successful.

While non-renormalisability of gravity becomes relevant at short distances, defining quantum observables including long-distance curvature effects has not been without problems either. In this direction, a consistent construction of scattering amplitudes in curved spacetime may provide a first hint on how to make progress in setting up a theory of Quantum Gravity.

Scattering amplitudes in flat spacetime have been applied to understand diverse aspects of gravity. A classic result is the construction of the Newtonian potential from the scattering amplitude in the tree-level Born approximation. The potential is then obtained by taking the Fourier transform in momentum space [18].

Treating gravity as an effective field theory, loop diagrams will give rise to two types of contributions to the gravitational potential: classical corrections due to the post-Newtonian expansion of GR, and purely quantum corrections. The implementation of extracting quantum corrections from the scattering amplitude of two gravitationally interacting massive particles has been explored in *e.g.* [19–22].

An alternative way to compute quantum corrections to the Newtonian potential is presented in [23–26]. When sources are moving slowly with respect to the speed of light and the gravitational field is weak, then the metric tensor can be expanded and expressed in terms of the so-called Bardeen variables. Computing the quantum-corrected equations of motion then allows to compute both post-Newtonian and quantum corrections to the gravitational potential.

The computation of scattering amplitudes in de Sitter spacetime has a long history. First attempts to construct such an amplitude gave divergent results [27]. This was attributed to the large-distance behaviour of the graviton propagator [28–30]. Later work showed that the divergence was a gauge artefact [31–37]. Different quantisation prescriptions [38–41] and invariance requirements [42, 43] were imposed, in order to obtain a finite form for the graviton propagator. In these works, the graviton propagator was obtained by projecting the two-point function into a complete basis for symmetric rank-2 tensor fields on the (Euclidean)

four-sphere [44] and then performing an analytic continuation.

There are a number of theoretical issues which seem to be special to de Sitter spacetime (see *e.g.* [45–58]). The choice of the vacuum (and of the coordinate system) can furnish distinct results for the propagator [59–62].[1] Moreover, in the analytical continuation of the propagator, different choices of the orientation of the branch cut can be made, producing different Lorentzian Green's functions. Finally, in de Sitter spacetime there exists a separate notion of spatial and temporal Fourier transformations, leading to subtleties in the representation in momentum space [73, 74].

Closely related to the de Sitter universe is the anti-de Sitter (AdS) spacetime. Here, the AdS/CFT correpondence allows one to compute correlators in the boundary CFT [75,76]. This was originally formulated for Euclidean signature [77–79]. In such a background, working in Euclidean signature usually does not lead to any restrictions since the results obtained in Euclidean AdS/CFT can be analytically continued to Minkowski space [80–84].

The struggle in the construction of observables related to scattering amplitudes also comprises the effort of computing the $S$-matrix in a de Sitter background [85–89]. The relation between the early and late time descriptions of particle states contains a great amount of dynamical information about the interacting theory. However, in de Sitter spacetime even one-particle states can decay and all particles are unstable [90–93]: there are no viable asymptotic states. This is related to the fact that there are no positive-definite energy-like conserved quantities. Furthermore, any observer will interact with a complete set of ingoing and outgoing states. Therefore, the $S$-matrix is not experimentally accessible to a single observer. Nonetheless, as we will demonstrate in this paper, it is perfectly possible to generalise Feynman diagrams to curved spacetime. We will adopt the assumption that the resulting object can still be interpreted as transition probabilities for scattering processes in de Sitter spacetime.

In this paper we present the amplitude of gravity-mediated scattering of two massive scalars in de Sitter spacetime. Novel in this work are two techniques. First, we represent vertices and propagators as differential operators, rather than through integral kernels. We will refer to this method as *operator method*. This generalises the Minkowski spacetime concept of momentum. Spacetime curvature is encoded in the noncommutativity of these operators, which we carefully take in to account using the framework developed in [94]. This comes with the additional advantage that the resulting expressions remain fully covariant.

Secondly, in order to convert the expression in terms of abstract differential operators into numerical quantities, we employ an expansion around infinite scalar masses. This is achieved by expanding in the dimensionless parameter $\mu = \frac{mc^2}{\hbar H}$, where $m$ is the particle's mass and $H = 68$ km/sec/Mpc is the Hubble constant.[2] We will refer to this expansion as the heavy-mass limit, and also goes under the name of de Sitter effective field theory [57]. In practice, such an expansion is well justified, with the electron mass compared to the Hubble constant being $m_e/H \approx 5 \cdot 10^{37}$ in natural units. Taking the limit $H \to 0$, we expect to retrieve the nonrelativistic limit in flat spacetime.

It is worth noticing that the expansion around $\mu = \infty$ has a slightly different status than the nonrelativistic limit in flat spacetime. In the latter case, a large-mass expansion is typically achieved by expanding around the dimensionless quantity $m/p$, or $p \ll m$. This explicitly breaks Lorentz invariance, since one explicitly refers to the frame-dependent spatial momentum. The expansion around $m/H = \infty$, however, is a Lorentz invariant procedure.

The rest of this paper is structured as follows. We first review the essentials of de Sitter spacetime in section 2, introducing conventions, coordinates and curvature relations. In section 3 we present the computation of the scattering amplitude of a tree-level graviton-

---

[1]During the quantisation procedure it is important to take into account the unitary irreducible representations of the de Sitter group (see [63–72] for a group theoretical approach).

[2]In the following, we convert to natural units by setting $\hbar = c = 1$.

mediated scalar-to-scalar scattering process. In section 4, we analyse this scattering amplitude in the heavy-mass limit. The main result of this section is the scattering potential presented in subsection 4.5. We end in section 5 with some concluding remarks and an outlook on extensions of this work. Explicit expressions for the propagator are given in Appendix A. Functional techniques dealing with the commutation of covariant derivatives with functions of the d'Alembertian are presented in Appendix B. Finally, Appendix C is dedicated to the computational details leading to the expression of the amplitude in the heavy-mass limit.

## 2 Essentials of de Sitter spacetime

In this section, we collect a few basic facts about de Sitter spacetime. The $d$-dimensional de Sitter spacetime is uniquely characterised as the maximally symmetric Lorentzian manifold with constant positive scalar curvature. It is the maximally symmetric solution of the vacuum Einstein equations including a positive cosmological constant $\Lambda$. The Ricci curvature tensor and the Ricci curvature scalar of de Sitter spacetime are given by [3]

$$R_{\mu\nu} = \frac{R}{d} g_{\mu\nu}, \qquad\qquad R = \frac{2d}{d-2}\Lambda, \qquad (1)$$

while the Weyl tensor vanishes. For $\Lambda > 0$, it is convenient to introduce the Hubble constant $H$:

$$R = d(d-1)H^2, \qquad\qquad H^2 = \frac{2}{(d-1)(d-2)}\Lambda. \qquad (2)$$

A negative cosmological constant, $\Lambda < 0$, corresponds to anti-de Sitter spacetime. Unless stated otherwise, we will assume throughout this paper that $\Lambda$ is positive.

De Sitter spacetime can be parameterised by various coordinate systems. Of particular importance in cosmology are comoving coordinates, which explicitly show that the universe is spatially homogeneous and isotropic. This property is encoded in the FLRW-like (or exponentially expanding) line element:

$$ds^2 = -dt^2 + e^{2Ht}(dx_1^2 + \cdots + dx_{d-1}^2). \qquad (3)$$

Introducing conformal time $\eta = -e^{-Ht}/H$, the metric is manifestly conformally flat:

$$ds^2 = \frac{1}{(H\eta)^2}\left(-d\eta^2 + dx_1^2 + \ldots + dx_{d-1}^2\right). \qquad (4)$$

Here, cosmic time $t$ runs from $-\infty$ to $+\infty$, while conformal time $\eta$ runs from $-\infty$ to 0, corresponding to past and future infinity, respectively.

## 3 The scattering amplitude functional

Having set the stage of the de Sitter manifold, we will now turn to the computation of a scattering amplitude. We will consider the graviton-mediated scattering of two particle species $\phi$ and $\chi$. In flat spacetime, at tree level this amplitude is described by a single $t$-channel Feynman diagram, depicted in Figure 1. In this section, we generalise this Feynman diagram to de Sitter spacetime by promoting propagators and vertices to noncommuting differential operators. Taking into account curvature contributions leads to the main result of this section, given in equations (18) and (20).

---

[3]We will use the mostly-plus convention for the metric. For the Riemann tensor, we use the convention $R_{\alpha\beta\gamma}{}^\delta = -\partial_\alpha \Gamma^\delta_{\beta\gamma} + \partial_\beta \Gamma^\delta_{\alpha\gamma} + \Gamma^\delta_{\beta\zeta}\Gamma^\zeta_{\alpha\gamma} - \Gamma^\delta_{\alpha\zeta}\Gamma^\zeta_{\beta\gamma}$. The Ricci tensor is given by $R_{\alpha\gamma} = R_{\alpha\beta\gamma}{}^\beta$.

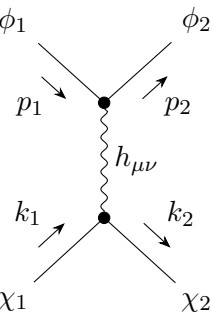

Figure 1: Tree-level graviton mediated scattering amplitude for the process $\phi_1\chi_1 \to \phi_2\chi_2$ ($t$-channel). Time flows from the left to the right.

## 3.1 Action

Before we describe how to compute the scattering amplitude in de Sitter spacetime, let us for concreteness first introduce the action. We will take an action that is a functional of the (dynamical) spacetime metric $\hat{g}$ and two scalar fields $\phi$ and $\chi$. Furthermore, we employ the background field method, depending on the background metric $\bar{g}$. The action is then of the following form:

$$S = S_{\text{grav}}[\hat{g}] + S_{\text{gf}}[\hat{g};\bar{g}] + S_{\text{sc}}[\phi,\hat{g}] + S_{\text{sc}}[\chi,\hat{g}].\tag{5}$$

Let us now specify each term in the action. For the gravitational action, we take the Einstein-Hilbert action

$$S_{\text{grav}}[\hat{g}] = \frac{1}{16\pi G_N} \int \mathrm{d}^d x \sqrt{-\hat{g}}\left(-2\Lambda + \hat{R}\right).\tag{6}$$

Here, we have denoted by a hat quantities defined with respect to the dynamical metric $\hat{g}$. At this stage, we do not specify the sign of the cosmological constant $\Lambda$. In order to obtain a well-defined graviton propagator, we add a de Donder type gauge fixing action [95]

$$S_{\text{gf}}[\hat{g};\bar{g}] = -\frac{1}{16\pi G_N}\frac{1}{2\alpha_{\text{gf}}} \int \mathrm{d}^d x \sqrt{-\bar{g}}\,\bar{g}^{\mu\nu}\mathcal{F}_\mu[\hat{g},\bar{g}]\mathcal{F}_\nu[\hat{g},\bar{g}],\tag{7}$$

where we defined the gauge fixing operator

$$\mathcal{F}_\mu[\hat{g},\bar{g}] = \delta_\mu^\beta \bar{g}^{\rho\sigma}\bar{\nabla}_\rho \hat{g}_{\sigma\beta} - \frac{1+\beta_{\text{gf}}}{d}\bar{g}^{\alpha\beta}\bar{\nabla}_\mu \hat{g}_{\alpha\beta}.\tag{8}$$

Since the Faddeev-Popov ghosts do not couple to the scalar field, they will not contribute to the tree-level scattering amplitude. We will therefore refrain from specifying their action here.

Finally, the matter sector will be given by minimally coupled, massive scalar fields $\phi$ and $\chi$, whose action reads

$$S_{\text{sc}}[\phi,\hat{g}] = -\frac{1}{2} \int \mathrm{d}^d x \sqrt{-\hat{g}}\,\phi\left(\hat{\Box} + m_\phi^2\right)\phi.\tag{9}$$

Here we have denoted by $\hat{\Box} = -\hat{g}^{\mu\nu}\hat{\nabla}_\mu\hat{\nabla}_\nu$ the covariant d'Alembertian.

## 3.2 Vertices and propagators

We will now describe how to compute the Feynman diagram from Figure 1 in the presence of a curved background metric $\bar{g}$ using the operator method. The Feynman diagram is given by

the contraction of the graviton propagator with two graviton-scalar-scalar three-point vertex operators. The vertices and propagators are obtained by taking functional derivatives with respect to the fields. This is done by expanding the action around a solution to the equation of motion. The equation of motion for the scalar fields is the Klein-Gordon equation,

$$\hat{\Box}\phi = -m_\phi^2 \phi\,, \qquad\qquad \hat{\Box}\chi = -m_\chi^2 \chi\,. \qquad (10)$$

Hence, $\phi = \chi = 0$ is a stationary point of the action, which we will use as expansion point for the action. The equation of motion for the metric is then Einstein's equation with cosmological constant, which is solved by the de Sitter metric. Therefore, we will write

$$\hat{g}_{\mu\nu} = \bar{g}_{\mu\nu} + h_{\mu\nu}\,, \qquad (11)$$

where $\bar{g}$ denotes the de Sitter metric.

We are now in the position to define the vertices and propagators. The three-point vertices are obtained by taking two derivatives with respect to the scalar field, and one with respect to the metric. After this, we set all fields to their background values. Hence, we obtain

$$\mathcal{T}^{(h\phi\phi)} = \frac{\delta^3 S}{\delta h\,\delta\phi\,\delta\phi}\bigg|_{\hat{g}=\bar{g}}\,, \qquad\qquad \mathcal{T}^{(h\chi\chi)} = \frac{\delta^3 S}{\delta h\,\delta\chi\,\delta\chi}\bigg|_{\hat{g}=\bar{g}}\,. \qquad (12)$$

The graviton propagator is defined to be the inverse of the two-point function:

$$\mathcal{G}^{(hh)} = \left[\frac{\delta^2 S}{\delta h\,\delta h}\right]^{-1}\bigg|_{\hat{g}=\bar{g}}\,. \qquad (13)$$

In order to make the inversion well-defined, it is necessary to include the gauge fixing action $S_{\text{gf}}$.

We note that the three-point vertices are formally linear operators acting on a graviton fluctuation and two scalar fluctuations, whereas the graviton propagator is a linear operator acting on two graviton fluctuations. From this point of view, the tree-level scattering amplitude is now given by contracting the graviton entries of the three-point vertices with the entries of the propagator. In order to make the amplitude easier to handle, we contract the other entries by on-shell scalar fields. This yields a number, given by the expression

$$\mathcal{A}[\chi_1,\chi_2,\phi_1,\phi_2] = \int \mathrm{d}^d x\,\sqrt{-\bar{g}}\,\left[\mathcal{T}^{(h\chi\chi)}[\chi_1,\chi_2]\right]^{\mu\nu}\left[\mathcal{G}^{(hh)}\right]_{\mu\nu}{}^{\rho\sigma}\left[\mathcal{T}^{(h\phi\phi)}[\phi_1,\phi_2]\right]_{\rho\sigma}\,. \qquad (14)$$

We will refer to this object as the amplitude functional. Note that the amplitude functional only contains objects in the de Sitter background. To ease the notation, we will therefore omit the bar to denote the background metric and its derived objects.

At this point, the following remark is in order. In our operator method setup, we will regard the vertices and propagator as differential operators, acting on scalar fields and metric fluctuations. This slightly formal point of view is in contrast to two common implementations of computing scattering amplitudes. First, in flat spacetime one has the Fourier transform at one's disposal. This allows to replace any derivatives with momenta reducing the scattering amplitude to a numerical quantity straightaway. In curved spacetime, such a Fourier transform is in general absent, and one has to take into account noncommutative derivatives. Secondly, the operator viewpoint has an advantage over a description in terms of position-space integral kernels. Regarding vertices and propagators as operators greatly simplifies composition and contraction of Feynman diagram elements, as opposed to computing lengthy integrals over de Sitter spacetime.

### 3.3 Computation of the propagator

Here we will give a brief description of how to compute the graviton propagator. By general covariance, the propagator can be written as a linear combination of functions of $\Box$ and $R$, multiplied by an appropriate tensor structure mapping a symmetric rank-two tensor to a symmetric rank-two tensor. Since we are working in de Sitter spacetime, we can choose a specific ordering of propagator functions and tensor structures. We choose to sort all propagator functions to the right, *i.e.*, we write

$$\mathcal{G}^{(hh)} = \sum_i \mathfrak{T}_i \mathcal{G}_i(\Box, R). \tag{15}$$

Here, $\mathfrak{T}_i$ are six tensor structures that span a basis of operators on rank-two tensors,

$$[\mathfrak{T}_i]_{\mu\nu}{}^{\rho\sigma} \in \left\{ \delta^{\rho}_{(\mu}\delta^{\sigma}_{\nu)}, g_{\mu\nu}g^{\rho\sigma}, g_{\mu\nu}\nabla^{(\rho}\nabla^{\sigma)}, \nabla_{(\mu}\nabla_{\nu)}g^{\rho\sigma}, \nabla_{(\mu}\delta^{(\rho}_{\nu)}\nabla^{\sigma)}, \nabla_{(\mu}\nabla_{\nu)}\nabla^{(\rho}\nabla^{\sigma)} \right\}, \tag{16}$$

and the functions $\mathcal{G}_i$ are the propagator functions that are to be computed.

We compute the propagator functions by demanding that

$$\left[ \frac{\delta^2 S}{\delta h \delta h} \right]_{\mu\nu}{}^{\rho\sigma} \left[ \mathcal{G}^{(hh)} \right]_{\rho\sigma}{}^{\alpha\beta} = \left[ \mathbb{1} \right]_{\mu\nu}{}^{\alpha\beta} = \delta^{\alpha}_{(\mu}\delta^{\beta}_{\nu)}. \tag{17}$$

We compute this in practice by acting with this equation onto a symmetric fluctuation $h_{\alpha\beta}$. Next, we sort all occurences of $\Box$ to the right, so that they act first on $h_{\mu\nu}$. In this way, together with the functions $\mathcal{G}_i$ they form differential operators with the appropriate index structure of an operator mapping symmetric two-tensors to symmetric two-tensors. Finally, we symmetrise all derivatives so that we map the result onto the basis (16). The propagator functions can then be read off; since their expressions are rather lengthy, we will not display them here, and refer to Appendix A and the attached notebook for additional details.

### 3.4 Computation of the amplitude functional

It remains to compute the amplitude functional by contracting the propagator with the three-point operators. We bring the amplitude to a standard form by sorting the contracted derivatives. In order to commute the contracted derivatives with the propagator functions, we employ the commutation techniques described in Appendix B. These techniques were developed in the context of affine gravity in [94]. Using integration by parts, we can manipulate on which scalar field each derivative acts. Finally, we can simplify the amplitude significantly by using the equation of motion (10) for the scalar fields.

With these methods, the amplitude functional can be brought to a standard form. We will require that the amplitude is manifestly symmetric in the external fields. Since there are at most four derivatives that are not contracted to a d'Alembertian, we infer that the amplitude takes the form

$$\mathcal{A} = \int \mathrm{d}^d x \sqrt{-g} \left( \chi_1 \chi_2 a_0(\Box) \phi_1 \phi_2 + T_{\mu\nu}[\chi_1, \chi_2] a_2(\Box) T^{\mu\nu}[\phi_1, \phi_2] \right), \tag{18}$$

where we have defined the symmetric tensor

$$T_{\mu\nu}[\phi_1, \phi_2] = \frac{1}{2} \left( (\nabla_\mu \nabla_\nu \phi_1)\phi_2 + \phi_1(\nabla_\mu \nabla_\nu \phi_2) \right). \tag{19}$$

Any other tensor structure can be brought to this form by partial integration and commuting derivatives.

The computation of the functions $a_0$ and $a_2$ is carried out with the tensor algebra package *xAct* [96–99]. The Mathematica notebook containing the computation is attached as ancillary file.

### 3.5 Result

We will now present the resulting operator functions $a_0$ and $a_2$. These constitute the first major result of this paper. They read

$$a_0(\square) = 2\pi G_N \left[ \square + \frac{8}{d} m_\phi^2 m_\chi^2 \left\{ \left( \square + \frac{2R}{d(d-1)} \right)^{-1} + \frac{2}{d-2} \left( \square - \frac{2R}{d} \right)^{-1} \right\} \right],$$

$$a_2(\square) = -16\pi G_N \left( \square + \frac{2R}{d(d-1)} \right)^{-1} . \tag{20}$$

With this result, the following observations can be made. Firstly, we remark that the computation for $\Lambda < 0$, and hence $R < 0$, gives the same result. Hence, the expressions (20) are valid in both de Sitter and anti-de Sitter spacetime. Specializing to dS, we can express $a_0$ and $a_2$ in terms of the Hubble parameter $H$:

$$a_0(\square) = 2\pi G_N \left[ \square + \frac{8}{d} m_\phi^2 m_\chi^2 \left\{ \left( \square + 2H^2 \right)^{-1} + \frac{2}{d-2} \left( \square - 2(d-1)H^2 \right)^{-1} \right\} \right],$$

$$a_2(\square) = -16\pi G_N \left( \square + 2H^2 \right)^{-1} . \tag{21}$$

Secondly, we note that $a_0(\square)$ and $a_2(\square)$ are non-local operators on de Sitter spacetime, acting on rank-0 and rank-2 tensors, respectively. The nontrivial behaviour is encoded in the operators of the form $(\square + z)^{-1}$ Using slightly sloppy terminology, we will refer to these as propagators, and to the parameters $z$ as graviton masses. It is good to keep in mind, however, that due to the background curvature any conclusions about whether the graviton is massive, massless or tachyonic cannot be drawn from the values of the graviton masses.

Thirdly, we observe that the amplitude is manifestly independent of the gauge fixing parameters $\alpha_{gf}$ and $\beta_{gf}$. This is in accordance with the expectation that the on-shell scattering amplitude is gauge invariant.

At this stage, it would be beneficial to compare our result (18) to existing computations, *e.g.* [32, 35, 39, 42, 56]. However these works do not employ an operator-based language as we have used here, making a detailed comparison difficult. In the next section, we attempt to make contact to these calculations by taking the heavy-mass limit.

Finally, we can perform a sanity check by taking the flat-spacetime limit by setting $R = 0$ and replacing all covariant derivatives by flat-spacetime derivatives. In this case the graviton masses $z_i$ become actual masses which vanish, making the masslessness of the graviton manifest. Furthermore, going to momentum space, and introducing the standard Mandelstam variables, we find in $d = 4$

$$A = \frac{2\pi G_N}{3} \frac{-\left( s^2 - 4su + u^2 + 2(m_\phi^2 - m_\chi^2)^2 \right) + (t - 2m_\phi^2)(t - 2m_\chi^2)}{t} . \tag{22}$$

This is in agreement with the standard result [100, 101].

## 4 Scattering amplitude and potential in the heavy-mass limit

We will now further analyse the amplitude functional. While the amplitude functional is given by abstract differential operators on de Sitter, we would like to be able to find a way to turn this into a concrete numerical expression. Problematic here is the fact that the propagators $(\square + z)^{-1}$ act on the product of two scalar fields. In order to deal with this, we will employ the heavy-mass limit, where we expand around the scalar masses $\mu = m/H = \infty$.

This section is structured as follows. First, after a brief review of quantisation of scalar fields, we will discuss the heavy-mass expansion of the scalar mode functions. We will then apply this expansion to the different objects in the amplitude functional. In particular, the heavy-mass expansion allows to set up differential equations describing the objects $(\Box + z)^{-1}\phi_1\phi_2$ and $(\Box + z)^{-1}T_{\mu\nu}$. The result can be summarized in an explicit expression for the tree-level scattering amplitude, which is presented in subsection 4.4. Fourier-transforming the scattering amplitude, we obtain the scattering potential in position space. Its construction and physical implications are discussed in subsection 4.5.

## 4.1 Scalar field quantisation and the heavy-mass limit

In this section, we will discuss the heavy-mass limit of the scalar fields $\phi$ and $\chi$. Before we dive into its derivation, let us briefly remind ourselves of the quantisation of scalar fields in Minkowski spacetime. The one-particle Hilbert space of a scalar field $\varphi$ with mass $m$ is constructed from solutions to the Klein-Gordon equation, whose solutions are plane waves parametrised by the spatial momentum $\vec{p}$:

$$\varphi_{\vec{p}} = a_+ e_{\vec{p}} + a_- \overline{e_{\vec{p}}}, \qquad\qquad e_{\vec{p}}(t,x) = e^{i(-\omega_p t + \vec{p}\cdot\vec{x})}. \tag{23}$$

Here, $a_\pm$ are constants and $\omega_p = \sqrt{p^2 + m^2}$, with $p = \|\vec{p}\|$ the standard Euclidean norm on spatial vectors, and denoted complex conjugation by a bar. The standard Minkowski vacuum is chosen by promoting $a_+$ to the annihilation operator $\hat{a}$ and $a_-$ to the creation operator $\hat{a}^\dagger$, corresponding to particles with positive energy $\omega_p$.

We point out two peculiar features of the Minkowski vacuum. First, the choice of mode functions $e_{\vec{p}}$ is the unique choice of solutions to the flat-spacetime Klein-Gordon equation such that the general solution can be written as the sum of $e_{\vec{p}}$ and its complex conjugate, as in (23). For the second property, we define the function

$$\tilde{\mathcal{E}}(m) := i\frac{\partial_t \varphi_{\vec{p}}}{\varphi_{\vec{p}}} = \omega_p \frac{a_+ e_{\vec{p}} - a_- \overline{e_{\vec{p}}}}{a_+ e_{\vec{p}} + a_- \overline{e_{\vec{p}}}}. \tag{24}$$

We now note that in general, $\tilde{\mathcal{E}}$ has an essential singularity at $m = \infty$, due to the appearance of $\omega_p$ in the complex exponential. The exception to this when $a_+ = 0$ or $a_- = 0$, i.e., in the Minkowski vacuum. In that case, we have

$$\tilde{\mathcal{E}}(m) = \pm\omega_p = \pm\left(m + \frac{p^2}{2m}\right) + \mathcal{O}(m^{-2}), \tag{25}$$

which is just the nonrelativistic expansion of the particle's energy. This analytic behaviour allows to consistently take the heavy-mass limit as an expansion around $m = \infty$.

We will now extend our analysis to de Sitter spacetime. The procedure runs completely analogous to flat-spacetime quantisation. For convenience, we will use conformal coordinates. First, we solve the Klein-Gordon equation in de Sitter spacetime. Again, the solutions are labeled by $\vec{p}$, which is now promoted to a comoving momentum. The solution is well-known [61]:

$$\varphi_{\vec{p}} = a_+ h_{\vec{p},\mu} + a_- \overline{h_{\vec{p},\mu}}, \qquad\qquad h_{\vec{p},\mu}(\eta,\vec{x}) = \eta^{\frac{d-1}{2}} H^{(1)}_{i\sqrt{\mu^2 - \left(\frac{d-1}{2}\right)^2}}(-p\eta) e^{i\vec{p}\cdot\vec{x}}. \tag{26}$$

Here we denoted by $H^{(1)}_\nu(z)$ the Hankel function of the first kind [102], and introduced the dimensionless quantity $\mu = m/H$. Note that for $z > 0$ and $\nu \in \mathbb{R}$, the Hankel function of the second kind is given by $H^{(2)}_{i\nu}(z) \propto \overline{H^{(1)}_{i\nu}(z)}$, which motivates the usual expansion in terms of

Hankel functions of first and second kind. We now quantise by promoting $a_+$ to the annihilation operator $\hat{a}$ and $a_-$ to the creation operator $\hat{a}^\dagger$. This prescription leads to the Bunch-Davies vacuum [103]. This is the canonical generalisation of the Minkowski vacuum, in the sense that $h_{\vec{p},\mu}$ reduces to a plane wave as $t \to -\infty$ (the Bunch-Davies boundary condition) [60,61,103], and in the limit $H \to 0$ [48].

In this work, we emphasise that the Bunch-Davies vacuum also generalises the Minkowski vacuum concerning the properties mentioned above: it is the unique choice of mode functions such that the coefficients of $a_\pm$ are each other's complex conjugates, and are the unique choice of modes that admit an expansion around $\mu = \infty$.[4] The first property is obvious; for the second property, we define the function

$$\mathcal{E}(\mu) = \mathrm{i}\frac{\partial_\eta \varphi_{\vec{p}}}{\varphi_{\vec{p}}}. \tag{27}$$

In order to compute the heavy-mass limit, we need to compute the expansion of $H_{\mathrm{i}\mu}^{(1)}(z)$ for large $\mu$. To this end, we note that the Hankel functions are defined to be solutions to Bessel's equation,

$$z^2 H''(z) + z H'(z) + (z^2 + \mu^2)H(z) = 0. \tag{28}$$

The expansion is computed by making the ansatz that the derivative of $H$ can be expressed in terms of $H$ itself, i.e.,

$$H'(z) = f(z)H(z). \tag{29}$$

Inserting this into Bessel's equation (28) gives the nonlinear equation

$$z^2 f'(z) + z^2 f(z)^2 + z f(z) + z^2 + \mu^2 = 0. \tag{30}$$

In analogy with the Minkowski vacuum, we will now look for a solution that has at most a simple pole in $\mu^{-1}$. Thus, we make the following ansatz for $f$:

$$f(z) = \mu \sum_{n \geq 0} f_n(z)\mu^{-n}. \tag{31}$$

Plugging this into (30) allows to solve the differential equation order by order in $\mu$. The first few equations read

$$\begin{aligned}
0 &= 1 + z^2 f_0^2, & 0 &= f_0 + 2z f_0 f_1 + z f_0', \\
0 &= z^2 + z^2 f_1^2 + 2z^2 f_0 f_2 + z^2 f_1', & 0 &= f_2 + 2z f_1 f_2 + 2z f_0 f_3 + z f_2'.
\end{aligned} \tag{32}$$

Solving these equations gives

$$f_\pm(z) = \pm\frac{\mathrm{i}\mu}{z} \pm \frac{\mathrm{i}z}{2\mu} - \frac{z}{2\mu^2} - \pm\frac{\mathrm{i}z(4+z^2)}{8\mu^3} + \mathcal{O}(\mu^{-4}). \tag{33}$$

In the accompanying notebook, we compute this expansion to higher orders. We note that for $z > 0$, the two solutions are each other's complex conjugate. We now have to show that this solution of (30) indeed gives an expansion of the Hankel functions. To this end, we expand $H_{\mathrm{i}\mu}^{(1)}(z)$ first around $z = 0$, and subsequently around $\mu = \infty$. This gives

$$H_{\mathrm{i}\mu}^{(1)}(z) = \frac{\mathrm{i}\mu}{z} + \mathcal{O}(z, \mu^{-1}). \tag{34}$$

---

[4]The requirement $m \gg H$ is a special case of the adiabatic approximation [104]. In general, adiabaticity assumes that the curvature of spacetime (here signified by $H$) is much smaller than the wavelength of the particle (given by the particle's mass). We refer to future work for a detailed discussion of the expansion in large mass in connection to the adiabatic approximation.

We see that this matches exactly the leading term of $f_+$; hence, we conclude that $f_+$ is indeed the expansion of $H_{i\mu}^{(1)}$. Plugging this into the definition of $\mathcal{E}$ in (27), we see that also in de Sitter space, $\mathcal{E}$ has a simple pole in $m$ if and only if $a_+$ or $a_-$ is zero. This motivates the definition of the Bunch-Davies vacuum as the de Sitter spacetime generalisation of the Minkowski vacuum.

We are now in the position to define the ingoing and outgoing states, and their respective heavy-mass limits. We associate an outgoing scalar field with the mode function $h_{\vec{p},\mu}$ and an ingoing scalar field with the mode function $\overline{h_{\vec{p},\mu}}$. Using the expansion (33), we can express their derivatives with respect to $\eta$ in terms of $h_{\vec{p},\mu}$:

$$\partial_\eta h_{\vec{p},\mu} = -i\mathcal{E}(\mu)h_{\vec{p},\mu} = \left[\frac{i\mu}{\eta} + \frac{d-1}{2\eta} + i\left(\frac{1}{2}p^2\eta^2 - \frac{(d-1)^2}{8\eta}\right)\frac{1}{\mu} + \mathcal{O}(\mu^{-2})\right]h_{\vec{p},\mu}. \tag{35}$$

With this definition at hand, we finally define the wave functions of the scalar fields to be:

$$\phi_1 = \overline{h_{\vec{p}_1,\mu_\phi}}, \qquad \phi_2 = h_{\vec{p}_2,\mu_\phi}, \qquad \chi_1 = \overline{h_{\vec{k}_1,\mu_\chi}}, \qquad \chi_2 = h_{\vec{k}_2,\mu_\chi}, \tag{36}$$

where we have defined the dimensionless masses $\mu_\varphi = m_\varphi/H$. This completes our definition of the scalar wave functions.

## 4.2 Heavy-mass expansion of the amplitude functional

With the heavy-mass expansion of the wave functions (35) at hand, we return to the amplitude functional (18). We will use the heavy-mass expansion to obtain concrete numerical quantities from this functional. To this end, we need to compute the heavy-mass expansions of $T_{\mu\nu}$, and of the action of $a_i(\square)$ on $T_{\mu\nu}$ and $\phi_1\phi_2$.

We will begin with the expansion of $T_{\mu\nu}$. It suffices to compute the expansion of all components in conformal coordinates. This gives

$$T_{00}[\chi_1,\chi_2] = \left[-\eta^{-2}\mu_\chi^2 - \frac{1}{2}(k_1^2 + k_2^2) + \frac{1}{2}(d^2-1)\eta^{-2} + \mathcal{O}(\mu_\chi^{-1})\right]\chi_1\chi_2,$$

$$T_{0i}[\chi_1,\chi_2] = \left[-\frac{1}{2}\eta^{-1}(k_{1,i} + k_{2,i})\mu_\chi + \frac{i}{4}(d+1)\eta^{-1}q_i + \mathcal{O}(\mu_\chi^{-1})\right]\chi_1\chi_2, \tag{37}$$

$$T_{ij}[\chi_1,\chi_2] = \left[-\frac{1}{2}(k_{1,i}k_{1,j} + k_{2,i}k_{2,j}) + \frac{1}{2}(d-1)\eta^{-2}\delta_{ij} + \mathcal{O}(\mu_\chi^{-1})\right]\chi_1\chi_2,$$

where Latin indices denote spatial coordinates. Here we have defined the comoving momentum transfer vector $\vec{q} = \vec{k}_2 - \vec{k}_1$, which is equal to $\vec{p}_1 - \vec{p}_2$ by momentum conservation. Note that only the (00)-component of $T_{\mu\nu}$ is of order $\mu_\chi^2$. Therefore, to leading order in $\mu_\chi$, there will only be a contribution from $\chi_1\chi_2 a_0(\square)\phi_1\phi_2$, and from $T_{00}[\chi_1,\chi_2]a_2(\square)T^{00}[\phi_1,\phi_2]$.

Before we start computing the heavy-mass expansion of $a_0(\square)\phi_1\phi_2$ and $a_2(\square)T_{\mu\nu}$, we have to show that the heavy-mass expansion exists, that is, the expansion in $\mu^{-1}$ contains at most finitely many positive powers of $\mu$. We will begin with showing that such an expansion exists for $\alpha(\square)\phi_1\phi_2$. We will assume that $\alpha$ has a Taylor series expansion; it then suffices to show that for $n$ sufficiently large, no new powers of $\mu$ are being generated.

This is shown by an inductive argument. First, we note that for $n = 0$, we have trivially

$$\square^n\phi_1\phi_2 = \phi_1\phi_2 \equiv \left[\alpha_0(\eta) + \mathcal{O}(\mu_\phi^{-1})\right]\phi_1\phi_2. \tag{38}$$

We now make the inductive assumption that (38) holds for $n$ d'Alembertians. Then acting with one more d'Alembertian gives

$$\square^{n+1}\phi_1\phi_2 = \square\alpha_0(\eta)\phi_1\phi_2 = \left[(q^2\eta^2\alpha_0 + d\eta\alpha_0' + \eta^2\alpha_0'')H^2 + \mathcal{O}(\mu_\phi^{-1})\right]\phi_1\phi_2. \tag{39}$$

Here, the prime denotes a derivative with respect to $\eta$. By induction, it follows that $\square^n \phi_1 \phi_2$ is $\mathcal{O}(\mu_\phi^0)\phi_1\phi_2$ for any $n$. Therefore, for any function $\alpha(\square)$ that admits an analytic expansion, $\alpha(\square)\phi_1\phi_2$ is $\mathcal{O}(\mu_\phi^0)\phi_1\phi_2$.

We will now prove in a similar manner that $\alpha(\square)T_{\mu\nu}$ has a well-defined expansion. However, we will now have to take care of the tensor structure of $T_{\mu\nu}$. From (37), we have seen that for $n = 0$, the following ansatz of $\square^n T_{\mu\nu}[\phi_1, \phi_2]$ holds:

$$
\begin{aligned}
\square^n T_{00}[\phi_1,\phi_2] &= \Big[ \alpha_{00}(\eta)\mu_\phi^2 + \mathcal{O}(\mu_\phi) \Big]\phi_1\phi_2\,, \\
\square^n T_{0i}[\phi_1,\phi_2] &= \Big[ \big( \alpha_1(\eta)p_{1,i} + \alpha_2(\eta)p_{2,i} \big)\mu_\phi^2 + \mathcal{O}(\mu_\phi) \Big]\phi_1\phi_2\,, \\
\square^n T_{ij}[\phi_1,\phi_2] &= \Big[ \Big( \alpha_{11}(\eta)p_{1,i}p_{1,j} + \alpha_{22}(\eta)p_{2,i}p_{2,j} + \alpha_{(12)}(\eta)(p_{1,i}p_{2,j} + p_{2,i}p_{1,j}) \\
&\qquad\qquad\qquad + \alpha_\delta(\eta)\delta_{ij} \Big)\mu_\phi^2 + \mathcal{O}(\mu_\phi) \Big]\phi_1\phi_2\,.
\end{aligned}
\tag{40}
$$

Here we have written down all possible tensor structures in de Sitter spacetime that are compatible with the symmetries of $T_{\mu\nu}$.

Let us now assume that for $n > 0$ the ansatz (40) holds. Then for $n + 1$, we obtain

$$
\begin{aligned}
\square^{n+1} T_{00}[\phi_1,\phi_2] = \Big[ \Big( &\eta^2 \alpha_{00}'' + (d+4)\eta\alpha_{00}' + q^2\eta^2\alpha_{00} - 2(d-1)\alpha_\delta \\
&+ 4i\eta(p_1^2 - \vec{p}_1 \cdot \vec{p}_2)\alpha_1 - 4i\eta(p_2^2 - \vec{p}_1 \cdot \vec{p}_2)\alpha_2 \\
&- 2p_1^2\alpha_{11} - 2p_2^2\alpha_{22} - 4\vec{p}_1 \cdot \vec{p}_2\alpha_{(12)} \Big)H^2\mu_\phi^2 + \mathcal{O}(\mu_\phi) \Big]\phi_1\phi_2\,.
\end{aligned}
\tag{41}
$$

The other components of $\square^{n+1} T_{\mu\nu}[\phi_1, \phi_2]$ also have leading order $\mu_\phi^2$. For simplicity, we will refrain from writing them down here; they can be found in the accompanying notebook. Thus, by induction it follows that $\square^n T_{\mu\nu}[\phi_1, \phi_2]$ is $\mathcal{O}(\mu_\phi^2)\phi_1\phi_2$ for any $n$. Similar to the scalar case, this can be analytically extended to any analytic function $\alpha$.

## 4.3 Expansion of the propagator

In this section, we will compute the action of the propagator on $\phi_1\phi_2$ and $T_{\mu\nu}[\phi_1, \phi_2]$. For generality, we will study the action of the operator $(\square + z)^{-1}$ for arbitrary mass parameter $z$. Later, we will specialise our result to the cases $z \in \{\}2H^2, (2 - 2d)H^2\}$, in accordance with (20). We will begin our analysis in the 0-derivative sector, *i.e.*, for $(\square + z)^{-1}\phi_1\phi_2$. After that, the action of $(\square + z)^{-1}$ on $T_{\mu\nu}$ is computed.

We begin our calculation with the observation that by the argument in subsection 4.2, the action of the action of the propagator $(\square + z)^{-1}$ on the product $\phi_1\phi_2$ is captured by the expansion

$$
(\square + z)^{-1}\phi_1\phi_2 = \Big[ G_0(\eta) + \mathcal{O}(\mu_\phi^{-1}) \Big]\phi_1\phi_2\,.
\tag{42}
$$

Next, we exploit the identity $(\square + z)(\square + z)^{-1} = \mathbb{1}$ by acting on (42) with $(\square + z)$. Distributing the d'Alembertian over $G_0$, we obtain the following second-order inhomogeneous differential equation:

$$
(\square + z)(\square + z)^{-1}\phi_1\phi_2 = \phi_1\phi_2 \qquad \Rightarrow \qquad \eta^2 G_0'' + d\eta G_0' + (q^2\eta^2 + \zeta)G_0 = \frac{1}{H^2}\,.
\tag{43}
$$

Here we have introduced the dimensionless mass parameter $\zeta = z/H$.

The space of solutions is two-dimensional, and the inhomogeneous solution reads

$$
G_0(\eta) = \frac{1}{\zeta H^2} \, {}_1\tilde{F}_2\left( 1; \frac{d+3}{4}, \frac{1}{4}\sqrt{(d-1)^2 - 4\zeta}; -\frac{q^2}{4} \right)\,,
\tag{44}
$$

where we have defined the proper momentum $\vec{\mathfrak{q}} := -\vec{q}\eta$ measured in units of $H$. In order to simplify the notation, we have written $_1\tilde{F}_2(a;b,c;z) := {}_1F_2(a;b-c,b+c;z)$ in terms of the generalised hypergeometric function $_1F_2$. The general solution is obtained by adding solutions to the homogenous equation. In general, these are non-analytic in $\eta$, and come with free parameters. At this stage, it is not clear how to fix these free parameters based on physical grounds; therefore, we will set them to zero.

We will now consider the action of the propagator on $T_{\mu\nu}$. As discussed in subsection 4.2, $(\Box + z)^{-1} T_{\mu\nu}[\phi_1, \phi_2]$ will be $\mathcal{O}(\mu_\phi^2)$ to leading order. This motivates the ansatz

$$
\begin{aligned}
(\Box + z)^{-1} T_{00}[\phi_1, \phi_2] &= \left[ G_{00}(\eta)\mu_\phi^2 + \mathcal{O}(\mu_\phi) \right] \phi_1 \phi_2 \,, \\
(\Box + z)^{-1} T_{0i}[\phi_1, \phi_2] &= \left[ \left( G_1(\eta)p_{1,i} + G_2(\eta)p_{2,i} \right)\mu_\phi^2 + \mathcal{O}(\mu_\phi) \right] \phi_1 \phi_2 \,, \\
(\Box + z)^{-1} T_{ij}[\phi_1, \phi_2] &= \left[ \left( G_{(12)}(\eta)(p_{1,i}p_{2,j} + p_{2,i}p_{1,j}) + G_\delta(\eta)\delta_{ij} \right. \right. \\
&\qquad \left. \left. + G_{11}(\eta)p_{1,i}p_{1,j} + G_{22}(\eta)p_{2,i}p_{2,j} \right)\mu_\phi^2 + \mathcal{O}(\mu_\phi) \right] \phi_1 \phi_2 \,.
\end{aligned}
\tag{45}
$$

Similar to the 0-derivative case, we now act with the operator $(\Box + z)$, and demand that this yields $T_{\mu\nu}[\phi_1, \phi_2]$. This gives a tensor equation, from which we can read off a set of coupled, second-order, inhomogeneous linear differential equations. The complete list of the equations is provided in Appendix C in (79).

The first step in solving these equations is to decouple. It turns out that this is only partially possible by linear transformations. Using the dimensionless proper momentum $\vec{\mathfrak{q}}$, we decouple the system into functions $g_i(\mathfrak{q}) = \sum_j a_{ij} G_j(\eta)$, where $i$ runs from 1 to 7. The partially decoupled equations are displayed in (80).

As noted in subsection 4.2, to leading order in $\mu_\chi$ the amplitude functional (18) depends on $(\Box + z)^{-1} T_{00}$ only; hence, we will be mainly interested in $G_{00}$. In decoupled variables, $G_{00}$ reads

$$
G_{00}(\eta) = \frac{q^2}{H^2} \left[ \left( \frac{1}{2} - \frac{1}{d} \right) g_1(\mathfrak{q}) + \frac{1}{2} g_2(\mathfrak{q}) + \frac{1}{d} g_6(\mathfrak{q}) \right].
\tag{46}
$$

For the sake of readability, we will reproduce the relevant differential equations here. The functions $g_1$ and $g_2$ occur in the following set of differential equations:

$$
\begin{aligned}
\mathfrak{q}^2 g_1'' + (d+4)\mathfrak{q} g_1' + (\mathfrak{q}^2 + \zeta + d + 2)g_1 &= \frac{1}{\mathfrak{q}^2} + d g_2 \,, \\
\mathfrak{q}^2 g_2'' + (d+4)\mathfrak{q} g_2' + (\mathfrak{q}^2 + \zeta + d)g_2 &= \frac{1}{\mathfrak{q}^2} + (d-2)g_1 + 4i\mathfrak{q} g_3 \,, \\
\mathfrak{q}^2 g_3'' + (d+4)\mathfrak{q} g_3' + (\mathfrak{q}^2 + \zeta + d)g_3 &= 4i\mathfrak{q} g_2 \,,
\end{aligned}
\tag{47}
$$

while $g_6$ is decoupled completely, and satisfies

$$
\mathfrak{q}^2 g_6'' + (d+4)\mathfrak{q} g_6' + \left( \mathfrak{q}^2 + \zeta + 2(d+1) \right) g_6 = \frac{1}{\mathfrak{q}^2}.
\tag{48}
$$

Again, we will be looking for the inhomogeneous solution to (47) and (48). The equation for $g_6$ is readily solved in terms of a generalised hypergeometric function:

$$
g_6(\mathfrak{q}) = \frac{1}{\zeta} \frac{1}{\mathfrak{q}^2} {}_1\tilde{F}_2 \left( 1; \frac{d+3}{4}, \frac{1}{4}\sqrt{(d-1)^2 - 4\zeta}; -\frac{\mathfrak{q}^2}{4} \right),
\tag{49}
$$

where we have used the same notation as in (44).

In order to solve the system (47), we will focus on the case $\zeta = 2$, since this value appears in the amplitude functional (18). We insert the following power series ansatz into the differential equations:

$$
g_i(\mathfrak{q}) = \mathfrak{q}^\nu \sum_{k \geq 0} a_{i,k} \mathfrak{q}^k.
\tag{50}
$$

Collecting powers of $\mathfrak{q}$ allows to set up a recursion relation for the coefficients $a_{i,k}$, which we can solve. The resulting power series can then be resummed in terms of generalised hypergeometric functions. Further details regarding solving the recursion relation are relegated to Appendix C and the accompanying notebook.

At this stage, a note about the mass parameter $\zeta = 2$ is in order. It turns out that this particular value of the mass parameter is special, causing the appearance of derivatives of the generalised hypergeometric functions with respect to their parameters. For $\zeta \neq 2$, these derivatives are absent. The solution at $\zeta = 2$ is *not* continuously connected to the solution for generic $\zeta$. We interpret this as a consequence of the masslessness of the graviton, which is signified in de Sitter spacetime by a non-zero mass parameter [105].

## 4.4 The amplitude in the heavy-mass limit

We are now in the position to construct the amplitude in the heavy mass limit. Carefully exploiting the expansions (37), (42) and (45), we find that the amplitude functional can be written as

$$\mathcal{A}[\chi_1, \chi_2, \phi_1, \phi_2] = \int \chi_1 \chi_2 \Big[ A(\mathfrak{q}) + \mathcal{O}(\mu_\chi, \mu_\phi) \Big] \phi_1 \phi_2 \,, \tag{51}$$

where

$$A(\mathfrak{q}) = 16\pi G_N \Big( -\eta^2 G_{00}^{\zeta=2} + \frac{1}{d} G_0^{\zeta=2} + \frac{2}{d(d-2)} G_0^{\zeta=-2(d-1)} \Big) H^4 \mu_\chi^2 \mu_\phi^2 \tag{52}$$

is the scattering amplitude. Plugging in the functions $G_0$ and $G_{00}$ gives

$$
\begin{aligned}
\frac{A(\mathfrak{q})}{8\pi G_N H^2 \mu_\chi^2 \mu_\phi^2} = &-\frac{(d-3)(119 + 23d + d^2 + d^3)}{12(d-1)(d+1)(d+5)} {}_0F_1\Big( \frac{d-3}{2}; -\frac{\mathfrak{q}^2}{4} \Big) \\
&+ \frac{(d-3)(-59 - 10d + d^2)}{12(d+1)(d+5)} {}_0F_1\Big( \frac{d-1}{2}; -\frac{\mathfrak{q}^2}{4} \Big) + {}_0F_1\Big( \frac{d+1}{2}; -\frac{\mathfrak{q}^2}{4} \Big) \\
&+ \frac{2}{(d-1)(d-2)} {}_1\tilde{F}_2\Big( 1; \frac{d-1}{4}, \nu_d; -\frac{\mathfrak{q}^2}{4} \Big) \\
&- \frac{1}{d-2} {}_1\tilde{F}_2\Big( 1; \frac{d+3}{4}, \nu_d; -\frac{\mathfrak{q}^2}{4} \Big) \\
&- \frac{2}{(d-1)(d-2)} {}_1\tilde{F}_2\Big( 2; \frac{d+3}{4}, \nu_d; -\frac{\mathfrak{q}^2}{4} \Big) \\
&+ \frac{\mathfrak{q}^2}{(d-1)^2} {}_1F_2^{(0;0,1;0)}\Big( 2; 2, \frac{d+1}{2}; -\frac{\mathfrak{q}^2}{4} \Big) \\
&- \frac{\mathfrak{q}^2}{(d-1)^2} {}_1F_2^{(0;1,0;0)}\Big( 2; 2, \frac{d+1}{2}; -\frac{\mathfrak{q}^2}{4} \Big) \,.
\end{aligned}
\tag{53}
$$

In this expression, we have denoted ${}_1F_2^{(0;n_a,n_b;0)}(2; a, b; z) = \partial_a^{n_a} \partial_b^{n_b} {}_1F_2(2; a, b; z)$ and defined the parameter $\nu_d = \frac{1}{4}\sqrt{(d-1)(d+7)}$.

For the analysis of the scattering amplitude, we will restrict ourselves to the case $d = 4$. A plot of the amplitude is depicted in Figure 2. There are several striking features of the scattering amplitude when compared to the nonrelativistic flat-spacetime amplitude, $A^{\text{flat}} \propto \frac{1}{q^2}$. First of all, the amplitude is bounded, in particular at $\mathfrak{q} = 0$, where it attains the value

$$\frac{A(\mathfrak{q} = 0)}{8\pi G_N H^2 \mu_\chi^2 \mu_\phi^2} = \frac{1}{6} \,. \tag{54}$$

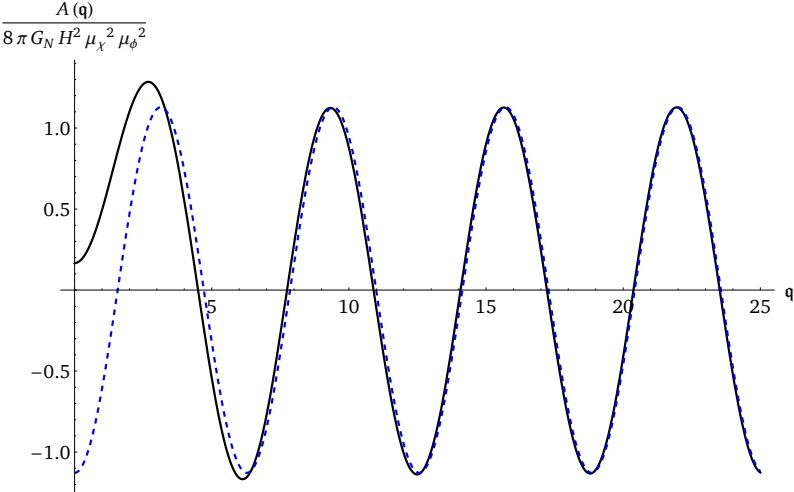

Figure 2: Plot of the scattering amplitude (53) in $d = 4$. We observe that, in contrast to the Minkowski-space $1/q^2$ amplitude, the de Sitter spacetime amplitude is bounded and oscillating. The dashed blue line denotes the asymptotic behaviour of $A(\mathfrak{q})$ calculated in (55).

This is in contrast to its Minkowski spacetime counterpart, which exhibits a pole at zero momentum transfer. We argue that the background curvature has acted as a natural infrared regulator, reminiscent of the exchange of a massive virtual particle.

The second striking feature displayed in Figure 2 is the oscillating behaviour of $A(\mathfrak{q})$. Performing an expansion around large $\mathfrak{q}$, we find

$$
\frac{A(\mathfrak{q})}{8\pi G_N H^2 \mu_\chi^2 \mu_\phi^2} \sim \left[ -\frac{397}{540} + \frac{2}{3}\log(2) + \frac{1}{3\sqrt{\pi}}\Gamma\left(\frac{3-\sqrt{33}}{4}\right)\Gamma\left(\frac{3+\sqrt{33}}{4}\right) \right]\cos(\mathfrak{q})
$$

$$
\approx -1.1283\cos(\mathfrak{q}). \tag{55}
$$

As is shown by the dashed blue line in Figure 2, this is already an excellent approximation to $A(\mathfrak{q})$ after a few oscillations. Keeping the comoving momentum $q$ fixed, $A(\mathfrak{q})$ can be regarded as oscillations in $\eta$. This phenomenon was also observed in [56], where it was related to particle production. In our context, it is interesting to interpret the absolute value of the amplitude as a probability density. A node at momentum $\mathfrak{q}_0$ then signifies that the exchange of a graviton with momentum $\mathfrak{q}_0$ is forbidden. Therefore, we conclude that graviton exchange is forbidden at discrete values of the transferred momentum. These momenta are equidistantly spaced with distance $\Delta\mathfrak{q} = 2\pi$.

We end this section with two remarks. Firstly, we note that the amplitude (53) is dimensionless in $d = 4$, which is consistent with power counting in flat spacetime. Secondly, we argue that the flat-spacetime limit of the amplitude $A(\mathfrak{q})$ is recovered correctly. Restoring dimensions by making the substitution $\mathfrak{q} \mapsto \mathfrak{q}/H$, we see that in the limit $H \to 0$, the period of the oscillations goes to zero, recovering a continuous spectrum of allowed exchanged momenta.

## 4.5 The scattering potential

Having obtained the scattering amplitude in momentum space, it is now interesting to convert this amplitude to position space. We will call this object the scattering potential, in analogy with the Born approximation in Minkowski-space quantum field theory. The computation of the potential is a straightforward generalisation of the flat-spacetime case.

The scattering potential is obtained by taking the Fourier transform of the amplitude. Preparing particle $\phi_1$ at initial position $\vec{x}_1$ and particle $\chi_1$ at $\vec{x}_2$, the transition probability amplitude is given by

$$V(\vec{x}_1, \vec{x}_2) = \frac{H}{2\mu_\phi \mu_\chi} \int \frac{d^{d-1}\vec{\mathfrak{k}}_1}{(2\pi)^{d-1}} \frac{d^{d-1}\vec{\mathfrak{p}}_1}{(2\pi)^{d-1}} e^{i\vec{\mathfrak{p}}_1 \cdot \vec{x}_1} e^{i\vec{\mathfrak{k}}_1 \cdot \vec{x}_2} A(\mathfrak{q}). \tag{56}$$

Here we have expressed the dimensionless proper momenta $\vec{\mathfrak{p}}_1 = -\eta \vec{p}_1$ and $\vec{\mathfrak{k}}_1 = -\eta \vec{k}_1$ in terms of comoving momenta in order to obtain an invariant result. We have also chosen the prefactor of the integral in such a way that the potential will be correctly normalised. The integral over $\vec{\mathfrak{k}}_1$ is easily seen to give a $\delta$-function; in order to perform the $\vec{\mathfrak{p}}_1$ integral, we shift the integration variable to obtain an integral over $\vec{\mathfrak{q}} = -\eta \vec{q}$. Writing the integral in spherical coordinates allows to perform the angular integrals. We thus obtain

$$V(\vec{x}_1, \vec{x}_2) = \delta^{d-1}(H\vec{x}_2) V(\mathfrak{r}), \tag{57}$$

$$V(\mathfrak{r}) = \frac{H}{2\mu_\phi \mu_\chi} \frac{2^{2-d} \pi^{\frac{1-d}{2}}}{\Gamma\left(\frac{d-1}{2}\right)} \int_0^\infty d\mathfrak{q}\, \mathfrak{q}^{d-2} {}_0F_1\left(\frac{d-1}{2}; -\frac{1}{4}\mathfrak{q}\mathfrak{r}\right) A(\mathfrak{q}), \tag{58}$$

where we have introduced the dimensionless proper distance $\mathfrak{r} = -\eta^{-1}\|\vec{x}_1 - \vec{x}_2\|$ between the two particles. Here we introduced the scattering potential $V(\mathfrak{r})$. As is clear from (58), the potential has dimension of energy in natural units, consistent with the classical notion of potential energy.

We will now present the scattering potential. The definition of the potential (58) has the shape of a Mellin transform of ${}_0F_1 \times {}_1F_2$. The evaluation of this transform was derived in [106]. Performing the integral, we find

$$V(\mathfrak{r}) = \begin{cases} V_{\mathfrak{r}<1}(\mathfrak{r}), & \mathfrak{r} < 1, \\ 0, & \mathfrak{r} > 1. \end{cases} \tag{59}$$

The function $V_{\mathfrak{r}<1}$ is rather complicated, and reads

$$\begin{aligned}
\frac{V_{\mathfrak{r}<1}(\mathfrak{r})}{\pi^{\frac{3-d}{2}} H^3 G_N \mu_\phi \mu_\chi} =& \frac{\Gamma\left(\frac{d-1}{2}\right)}{d-1}\left[-8\mathfrak{r}^{3-d} + 2\left(-1 + 4\mathfrak{r}^2 + (d+\mathfrak{r}^2 - d\mathfrak{r}^2)^2\right)\right]\frac{1}{(\mathfrak{r}^2-1)^2} \\
&+ \frac{2(d-1)\Gamma\left(\frac{d-3}{2}\right)}{d-2}\mathfrak{r}^{3-d} {}_2\tilde{F}_1\left(\frac{5-d}{4}, \nu_d; \frac{5-d}{2}; \mathfrak{r}^2\right) \\
&- \frac{8\Gamma\left(\frac{d-3}{2}\right)}{d-1}\mathfrak{r}^{3-d} {}_2\tilde{F}_1\left(\frac{9-d}{4}, \nu_d; \frac{5-d}{2}; \mathfrak{r}^2\right) \\
&- 4\Gamma\left(\frac{d-5}{2}\right)\mathfrak{r}^{5-d} {}_2\tilde{F}_1\left(\frac{9-d}{4}, \nu_d; \frac{7-d}{2}; \mathfrak{r}^2\right) \\
&- \frac{8\Gamma\left(\frac{5-d}{2}\right)}{(d-2)(d-1)}\frac{\Gamma\left(\frac{d+3}{4}-\nu_d\right)\Gamma\left(\frac{d+3}{4}+\nu_d\right)}{\Gamma\left(\frac{5-d}{4}-\nu_d\right)\Gamma\left(\frac{5-d}{4}+\nu_d\right)} {}_2\tilde{F}_1\left(\frac{d-1}{4}, \nu_d; \frac{d-3}{2}; \mathfrak{r}^2\right) \\
&- \frac{4\Gamma\left(\frac{3-d}{2}\right)}{d-2}\frac{\Gamma\left(\frac{d+3}{4}-\nu_d\right)\Gamma\left(\frac{d+3}{4}+\nu_d\right)}{\Gamma\left(\frac{5-d}{4}-\nu_d\right)\Gamma\left(\frac{5-d}{4}+\nu_d\right)} {}_2\tilde{F}_1\left(\frac{d-1}{4}, \nu_d; \frac{d-1}{2}; \mathfrak{r}^2\right) \\
&- \frac{4\Gamma\left(\frac{1-d}{2}\right)}{d-2}\frac{\Gamma\left(\frac{d-1}{4}-\nu_d\right)\Gamma\left(\frac{d-1}{4}+\nu_d\right)}{\Gamma\left(\frac{1-d}{4}-\nu_d\right)\Gamma\left(\frac{1-d}{4}+\nu_d\right)} {}_2\tilde{F}_1\left(\frac{d+3}{4}, \nu_d; \frac{d-1}{2}; \mathfrak{r}^2\right).
\end{aligned} \tag{60}$$

Here we introduced the shorthand notation ${}_2\tilde{F}_1(a, b; c; z) = {}_2F_1(a-b, a+b; c; z)$.

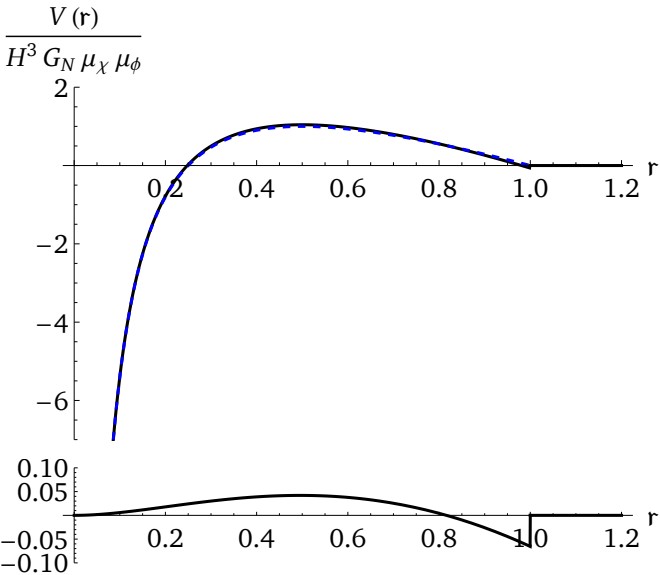

Figure 3: Plot of the large-mass limit tree-level scattering potential in de Sitter space-time. Top panel: the potential $V$ is shown by a solid black line. The dashed blue line shows the approximation $V_{app}$ from (61). Bottom panel: the residual values $V - V_{app}$. We observe that the residual values are $\lesssim 0.1$, indicating that the approximation captures the behaviour of $V$ well.

Before we start analysing the properties of the potential, let us remark that the appearance of $_2F_1$ hypergeometric functions is in accordance with existing computations [45, 107]. This is a signature of the virtual graviton propagator, which written as an integration kernel in position space is proportional to $_2F_1$.

In the remainder of this paragraph, we will set $d = 4$. We can then plot $V(\mathfrak{r})$, which is given in Figure 3. The plot shows several remarkable features. First, we expand around $\mathfrak{r} = 0$. This gives the approximation

$$\frac{V(\mathfrak{r})}{H^3 G_N \mu_\phi \mu_\chi} \sim V_{app}(\mathfrak{r}) = -\frac{1}{\mathfrak{r}} + 5 - 4\mathfrak{r}. \tag{61}$$

This approximation is shown by a dashed blue line in Figure 3. As can be seen from the residual plot, this captures the behaviour of $V(\mathfrak{r})$ very well. From the (61), we see that the leading-order term is exactly the flat-spacetime Newtonian potential. This is in accordance with the observation that at small distances, the background curvature of spacetime can be neglected.

Classically, the Newtonian force is given by the derivative of the potential. This gives

$$F \propto -\frac{V'(\mathfrak{r})}{H^3 G_N \mu_\phi \mu_\chi} \sim -\frac{1}{\mathfrak{r}^2} + 4. \tag{62}$$

Hence, we conclude that the correction to the Newtonian potential gives rise to an approximately constant repulsive force. This is in agreement with the interpretation of the expansion of the universe screening the attractive gravitational force between the two particles. As a matter of fact, we observe that the expanding force dominates at large distances, demonstrated by a maximum of the potential at $\mathfrak{r}_{max} \approx 0.4998$, where the potential takes the value $V(\mathfrak{r}_{max}) \approx 1.0420 H^3 G_N \mu_\phi \mu_\chi$. To large extent, this is dominated by the leading-order terms in (61), which yields an approximate maximum at $\mathfrak{r} = \frac{1}{2}$ at $V_{app}(1/2) = 1 \cdot H^3 G_N \mu_\phi \mu_\chi$.

We observe that the potential remains finite as it approaches $\mathfrak{r} = 1$ from below. The limit is given by

$$\frac{V(\mathfrak{r}=1)}{H^3 G_N \mu_\phi \mu_\chi} = \frac{5}{2} + \frac{2\pi^{3/2}\sec\left(\frac{\sqrt{33}\pi}{2}\right)}{\Gamma\left(\frac{1}{4}(1-\sqrt{33})\right)\Gamma\left(\frac{1}{4}(1+\sqrt{33})\right)} \approx -0.06553\,. \tag{63}$$

We remark that these potential energies are measured in terms of the Hubble energy, which is given by $\hbar^2 H^3 G_N/c^5 \approx 2\cdot 10^{-155}$ eV. It is obvious that these energy differences are too tiny to be measured in earth-based experiments.

For $\mathfrak{r} > 1$, the potential vanishes identically. This can be naturally explained as a manifestation of the de Sitter horizon: particles separated at super-Hubble scales cannot causally interact. Therefore,the scattering potential explicitly encodes the causal structure of de Sitter spacetime.

We close the discussion of the potential with the following remark. Since the potential has bounded support, any spacetime integral over the potential will be cut off at $\mathfrak{r} = 1$. Hence, it is not plagued by an IR divergence. This is in contrast to the Minkowski-spacetime potential, where integrals over the amplitude typically exhibit a logarithmic divergence. This is dual to the finite limit of $A(\mathfrak{q})$ as $\mathfrak{q}$ approaches zero. We argue here that the curvature of the background spacetime plays the role of an infrared regulator.

## 5 Conclusion and outlook

In this paper, we have constructed the amplitude and potential of a scalar-to-scalar scattering process in a de Sitter spacetime, mediated by gravitons, in the tree-level approximation. Novel in this work is the application of operator methods to construct the graviton propagator and its contraction with the scalar vertices in a fully covariant way, resulting in a covariant amplitude functional. The amplitude functional is explicitly gauge-independent, and reduces to the well-known flat-spacetime scattering amplitude in the limit $R \to 0$. Interesting is the appearance of operators of the form $(\Box + z)^{-1}$, where $z \in \{2H^2, -2(d-1)H^2\}$, resembling massive propagators. This hints towards a regularisation of the graviton propagator in the infrared regime, opposed to the $1/q^2$ divergence in Minkowski spacetime. Furthermore, one of the masses comes with a negative sign, which raises questions about the tachyonic nature of the graviton. However, as we have shown in this paper, the nontrivial curvature effects conspire to give a fully regular IR behaviour.

In the second part of this paper, we extracted the tree-level scattering amplitude and potential from the abstract amplitude functional. Central in this computation is the application of the heavy-mass limit of the scalar fields, allowing us to obtain the distribution of the graviton propagator over the product of two scalar fields. The heavy-mass limit is well-justified due to the observed small value of the cosmological constant compared to *e.g.* the electron mass.

From the scattering amplitude, we construct the following compelling physical picture. First, we observe that the amplitude has a finite limit as the transferred graviton momentum goes to zero, $\mathfrak{q} \to 0$. This is in contrast to the Minkowski-spacetime amplitude, which exhibits an IR divergence due to the masslessness of the graviton. This is evidence for the regularisation of the graviton propagator due to corrections from the curved background. Second, a striking feature of the scattering amplitude is its oscillating behaviour as a function of $\mathfrak{q}$. Hence, the scattering probability vanishes for discrete values of the graviton momentum $\mathfrak{q}$. This discrete behaviour is reminiscent of the discrete transition probabilities observed with particles in a box. In the cosmological context, this would translate to graviton exchange being bounded by the Hubble volume.

These physical features also recur in the scattering potential. For small values of the radius, this reproduces the Newtonian $1/r$ potential. At larger distances, modifications due to the background curvature induce corrections to the potential, which to leading order can be captured by a constant repulsive force. For distances of approximately half the Hubble radius, this force dominates the attractive Newtonian force, such that particles are repelled. This is in line with the interpretation of the de Sitter universe as an expanding spacetime. As the radius approaches the de Sitter horizon, the potential reaches a finite value, before vanishing identically for $\mathfrak{r} > 1$. Hence, the potential has bounded support while remaining bounded from above. This is the position-space analogue of the finite IR properties of the amplitude. Finally, the vanishing of the potential at super-Hubble distances is completely in agreement with the causal properties of the de Sitter spacetime, which forbid any causal interaction of particles beyond the horizon.

At this point, we look ahead to further implications of the heavy-mass limit. This approximation differs significantly from the usual derivation of the Newtonian potential in GR, where the "test particle" is assumed to be light, such that the backreaction of the geometry can be neglected. This explains the fact that only to a limited extent, the scattering potential (59) reproduces the classical GR potential [108, 109], given by $V_{\text{eff}}(r) = -\frac{G_N m}{r} - \frac{1}{6}\Lambda r^2$. Instead, taking the scattering amplitude as a tool to reconstruct the backreaction of matter on spacetime, we conclude that the amplitude probes an effective geometry where both a cosmological constant $\Lambda$ and two heavy (black-hole) masses are present. For such a geometry, no explicit solutions of Einstein's equations are known.

This conclusion may also have profound consequences for the properties for the de Sitter horizon. This famously has a temperature [110, 111], comparable to the Bekenstein-Hawking temperature of a black hole [107, 112]. The present system allows to probe the horizon properties of a more general geometry with two masses inside. This may shed light on the outstanding problem of defining a horizon temperature in a Schwarzschild-de Sitter spacetime [113]. Particularly intriguing in this context is the discontinuity of the potential at $\mathfrak{r} = 1$. In analogy with classical electrodynamics, the discontinuity may be interpreted as a surface energy density situated at the horizon. In a semi-classical interpretation, the discontinuity would allow for tunneling of particles. It would be highly interesting to study both explanations, and try to make contact with a thermodynamic description of the de Sitter horizon.

A different line to follow is to connect the novel formalism introduced here to more established methods to compute de Sitter correlators [45–55, 57, 58]. The approach followed in this work differs on several crucial points which makes it difficult to compare the two strategies. First, we employ an operator-based scheme, which allows us to work in a manifestly covariant way. Second, in order to turn the operator expressions into a concrete amplitude, it was necessary to use the heavy-mass limit. A solid understanding of the relation between our approach and dS correlator methods would be very helpful to grasp the nature of the objects that we are studying.

In future work, the scattering amplitude presented here can be extended in the following ways. The first step is to compute higher-order corrections in the heavy-mass expansion. We already anticipate that the next-to-leading order in $m_\chi$ can easily be computed using (37). This will involve solving the full system of equations (79). An interesting consequence is that this will likely introduce angular dependence into the scattering amplitude, which may give information about the spin of the transferred graviton. Taking into account the next-to-leading order in $m_\phi$ will prove to be more difficult, since this requires computing the action of the propagator on $T_{\mu\nu}$ to the next order. At this moment, it is unclear whether the resulting system of equations is tractable. Furthermore, a word of caution is in place here. We have not investigated the convergence properties of the expansion in large masses. This leaves open the possibility that the expansion holds only asymptotically. Hence, in the large-distance regime,

where subleading contributions may become relevant, this has to be taken into account.

A second extension is to include non-minimal couplings. The inclusion of scalar-gravity interactions such as an $R\phi\phi$-coupling, and modifications to the graviton propagator, given by an $R^2$ or a Weyl-square $C^2$ term in the action, is completely straightforward. Adding higher-derivative terms to the gravitational action leads to Stelle gravity [114,115]. This is known to be perturbatively renormalisable, but plagued by a spin-2 ghost due to the presence of a pole with negative mass in the propagator. Taking into account corrections due to the background curvature may alleviate this problem.

Furthermore, it is especially tantalising in this setting that the inclusion of non-minimal interactions may lead to a modified potential at an intermediate length scale, set by the size of the non-minimal couplings. Tuning the size of these couplings to, say, galactic scales may lead to a MOND-like scenario [116], where the coupling of non-minimal gravitational couplings provide an elegant explanation for dark matter.

Third, a necessary challenge to be taken up is the inclusion of quantum corrections. A convenient scheme to capture these is the form factor formalism [101], which can be extended to curved spacetime. Starting point of this formalism is the quantum effective action, which encodes fully-dressed scattering amplitudes by means of tree-level diagrams. The momentum-dependence of the effective vertices and propagators are parameterised by form factors, which appear in curved spacetime as functions of covariant derivatives. The form factor language allows for a uniform description of scattering processes for a large number of models of quantum gravity. By specifying the form factors, one can include quantum corrections encoded in loop diagrams, such as those computed in a Minkowski-spacetime background, or one can include quantum corrections obtained using non-perturbative techniques [117]. Choosing different form factors, the effective action can be reduced to the string theory low-energy effective action [118,119], it can describe non-local ghost-free gravity [120], allows to study non-perturbative unitarity properties [121] and permits models mimicking the Veneziano amplitude [122]. In this context, it would be highly illuminating to compute the scattering amplitude dressed by form factors, and the specific values of the form factors given by loop corrections in a de Sitter background.

Finally, the formalism developed in this work can be extended to other curved backgrounds. Of particular interest in cosmology would be to extend the de Sitter metric to a generic FLRW background. In order to make the computation tractable, one could assume that the FLRW corrections to de Sitter spacetime are small, for example in the slow-roll approximation. In a different direction, a treatment of scattering in a Schwarzschild black-hole background would be relevant to a number of open problems in quantum field theory. Generalising our methods to a black hole background may shed light on the role of unitarity, and the process of black hole evaporation.

In conclusion, it goes without saying that scattering amplitudes in curved spacetime give a fascinating outlook on gravitational observables. They provide set of methods to understand the quantum nature of spacetime and its contents. We believe that the techniques developed in this work will be an important tool in this toolbox.

# Acknowledgements

We are grateful to the Theoretical Cosmology group at Tokyo Institute for Technology and the Quantum Gravity group at Radboud University Nijmegen for illuminating discussions in the course of this work. We would like to thank Martin Reuter, Benjamin Knorr and K. Sravan Kumar for helpful comments on the manuscript.

**Funding information** This work is partially supported by DFG Grant No. RE 793/8-1.

## A  The propagator operator in a curved background

In this section, we will provide the explicit expressions for the propagator $\mathcal{G}^{(hh)}$. The propagator is of the form

$$\mathcal{G}^{(hh)} = \sum_i \mathfrak{T}_i \mathcal{G}_i(\square, R). \tag{64}$$

Here the index $i = 1, \ldots, 6$ runs over the set of tensor structures given in (16). We will now present the functions $\mathcal{G}_i(\square, R)$. For simplicity, we will restrict ourselves to the case $d = 4$. Nontrivial in this computation are the gauge parameters $\alpha_{\text{gf}}$ and $\beta_{\text{gf}}$, which we will keep arbitrary. The functions $\mathcal{G}_i$ are given by

$$
\begin{aligned}
\frac{3}{32\pi G_N}\mathcal{G}_1(\square,R) = & -(\alpha_{\text{gf}}-1)RG\left(\tfrac{1}{6}\right)^2 - 2\alpha_{\text{gf}}G\left(\tfrac{1}{6}\right) + 2\frac{\alpha_{\text{gf}}-\beta_{\text{gf}}}{\beta_{\text{gf}}-3}RG\left(\tfrac{2}{3}+\tfrac{1}{\beta_{\text{gf}}-3}\right)^2 \\
& + 2\left(\alpha_{\text{gf}}-3\right)G\left(\tfrac{2}{3}+\tfrac{1}{\beta_{\text{gf}}-3}\right),
\end{aligned}
$$

$$
\begin{aligned}
\frac{1}{16\pi G_N}\mathcal{G}_2(\square,R) = & \tfrac{1}{6}(\alpha_{\text{gf}}-1)RG\left(\tfrac{1}{6}\right)^2 + \tfrac{1}{3}(\alpha_{\text{gf}}-1)G\left(\tfrac{1}{6}\right) - \tfrac{1}{3}\frac{\alpha_{\text{gf}}-\beta_{\text{gf}}}{\beta_{\text{gf}}-3}G\left(\tfrac{2}{3}+\tfrac{1}{\beta_{\text{gf}}-3}\right)^2 \\
& - \frac{\alpha_{\text{gf}}-7-\frac{6}{\beta_{\text{gf}}-3}}{3}G\left(\tfrac{2}{3}+\tfrac{1}{\beta_{\text{gf}}-3}\right) - \frac{2}{\beta_{\text{gf}}-3}G\left(\tfrac{1}{\beta_{\text{gf}}-3}\right),
\end{aligned}
$$

$$
\begin{aligned}
\frac{9}{128\pi G_N}\mathcal{G}_3(\square,R) = & (\alpha_{\text{gf}}-1)G\left(\tfrac{1}{6}\right)^2 - \frac{2\alpha_{\text{gf}}-15}{R}G\left(\tfrac{1}{6}\right) + 2\frac{\alpha_{\text{gf}}-\beta_{\text{gf}}}{\beta_{\text{gf}}-3}G\left(\tfrac{2}{3}+\tfrac{1}{\beta_{\text{gf}}-3}\right)^2 \\
& + \frac{(2\alpha_{\text{gf}}-15)}{R}G\left(\tfrac{2}{3}+\tfrac{1}{\beta_{\text{gf}}-3}\right),
\end{aligned}
$$

$$
\begin{aligned}
\frac{1}{128\pi G_N}\mathcal{G}_4(\square,R) = & -\tfrac{2}{9}(\alpha_{\text{gf}}-1)G\left(\tfrac{1}{6}\right)^2 + \tfrac{1}{9}\frac{4\alpha_{\text{gf}}-3}{R}G\left(\tfrac{1}{6}\right) \\
& - \tfrac{1}{9}\frac{(\alpha_{\text{gf}}-\beta_{\text{gf}})(4\beta_{\text{gf}}-3)}{(\beta_{\text{gf}}-3)^2}G\left(\tfrac{2}{3}+\tfrac{1}{\beta_{\text{gf}}-3}\right)^2 \\
& - \tfrac{1}{9}\frac{4\alpha_{\text{gf}}-3}{R}G\left(\tfrac{2}{3}+\tfrac{1}{\beta_{\text{gf}}-3}\right) + \frac{\alpha_{\text{gf}}-\beta_{\text{gf}}}{(\beta_{\text{gf}}-3)^2}G\left(\tfrac{1}{\beta_{\text{gf}}-3}\right)^2,
\end{aligned}
$$

$$
\begin{aligned}
\frac{9}{128\pi G_N}\mathcal{G}_5(\square,R) = & -(\alpha_{\text{gf}}-1)G\left(\tfrac{1}{6}\right)^2 + 11\frac{\alpha_{\text{gf}}-3}{R}G\left(\tfrac{1}{6}\right) - 11\frac{\alpha_{\text{gf}}-\beta_{\text{gf}}}{\beta_{\text{gf}}-3}G\left(\tfrac{2}{3}+\tfrac{1}{\beta_{\text{gf}}-3}\right)^2 \\
& - 11\frac{\alpha_{\text{gf}}-3}{R}G\left(\tfrac{2}{3}+\tfrac{1}{\beta_{\text{gf}}-3}\right),
\end{aligned}
$$

$$
\begin{aligned}
\frac{1}{256\pi G_N}\mathcal{G}_6(\square,R) = & \frac{\alpha_{\text{gf}}-1}{R}G\left(\tfrac{1}{6}\right)^2 - 2\frac{\alpha_{\text{gf}}-3}{R^2}G\left(\tfrac{1}{6}\right) + 2\frac{\alpha_{\text{gf}}-\beta_{\text{gf}}}{\beta_{\text{gf}}-3}G\left(\tfrac{2}{3}+\tfrac{1}{\beta_{\text{gf}}-3}\right)^2 \\
& + 2\frac{\alpha_{\text{gf}}-3}{R^2}G\left(\tfrac{2}{3}+\tfrac{1}{\beta_{\text{gf}}-3}\right).
\end{aligned}
\tag{65}
$$

Here, we defined the shorthand notation $G_\nu = (\square + \nu R)^{-1}$. In the De Donder gauge, $\alpha_{\text{gf}} = 1$, $\beta_{\text{gf}} = \frac{d}{2}-1$, this reduces to

$$\frac{\mathcal{G}_1(\square,R)}{16\pi G_N} = -4G\left(\tfrac{1}{6}\right), \qquad \frac{\mathcal{G}_2(\square,R)}{16\pi G_N} = G\left(-\tfrac{1}{2}\right) + G\left(\tfrac{1}{6}\right), \qquad \mathcal{G}_3 = \mathcal{G}_4 = \mathcal{G}_5 = \mathcal{G}_6 = 0. \tag{66}$$

Further details, including the expressions for the propagator in arbitrary dimension, can be found in the attached notebook.

# B    Commutation relations in constantly-curved spaces

In order to compute the propagator and to bring the amplitude functional to a canonical form, we need to compute the commutator of the operator $f(\Box)$ with covariant derivatives in a de Sitter background. The following appendix directly follows [94]; here the formalism has been adapted to metric gravity.

We want to obtain a formula of the form

$$f(\Box)\nabla_\alpha \mathbf{X} = \nabla_\alpha f(\Box)\mathbf{X} + \dots , \tag{67}$$

where $f$ is an arbitrary function and $\mathbf{X}$ is a tensor of rank $(0, n)$,

$$\mathbf{X} = X_{\mu_1 \cdots \mu_n} . \tag{68}$$

To derive an equation of the form (67), we employ a standard trick: we express the function $f$ as an inverse Laplace transform, so that we can use the Baker-Campbell-Hausdorff formula,

$$f(\Box)\nabla_\alpha \mathbf{X} = \int_0^\infty \mathrm{d}s\, \tilde{f}(s)\, \mathrm{e}^{-s\Box}\nabla_\alpha \mathbf{X} = \int_0^\infty \mathrm{d}s\, \tilde{f}(s) \sum_{\ell \geq 0} \frac{(-s)^\ell}{\ell!} [\Box, \nabla_\alpha]_\ell\, \mathrm{e}^{-s\Box}\mathbf{X}. \tag{69}$$

Here we used the multi-commutator, which is defined recursively by

$$[A, B]_n = [A, [A, B]_{n-1}], \qquad\qquad [A, B]_0 = B . \tag{70}$$

In order to compute the multi-commutator, let us first calculate the standard commutator. Assuming that we are working in de Sitter spacetime, we find

$$\begin{aligned}
[\Box, \nabla_\alpha] X_{\mu_1 \cdots \mu_n} &= -\frac{R}{d}\nabla_\alpha X_{\mu_1 \cdots \mu_n} + \frac{2}{d-1}\frac{R}{d}\sum_{k=1}^n \Big[\bar{g}_{\alpha\mu_k}\nabla^\beta X_{\mu_1 \cdots \mu_{k-1}\beta\mu_{k+1}\cdots\mu_n} \\
&\qquad\qquad\qquad\qquad\qquad\qquad - \nabla_{\mu_k} X_{\mu_1 \cdots \mu_{k-1}\alpha\mu_{k+1}\cdots\mu_n}\Big] \\
&\equiv C_{\alpha\mu_1\cdots\mu_n}{}^{\beta\,\nu_1\cdots\nu_n}\nabla_\beta X_{\nu_1\cdots\nu_n} .
\end{aligned} \tag{71}$$

We observe that the commutator is a multiplication with a covariantly constant tensor $\mathbf{C}$, so that structurally we find

$$[\Box, \nabla]_k \mathbf{X} = \mathbf{C}^k \nabla \mathbf{X}. \tag{72}$$

We can plug this back into the original equation (69), so that we are left with

$$f(\Box)\nabla\mathbf{X} = \int_0^\infty \mathrm{d}s\, \tilde{f}(s)\, \mathrm{Texp}[-s\mathbf{C}]\,\nabla\mathrm{e}^{-s\Box}\mathbf{X}, \tag{73}$$

where Texp is the tensor exponential. This can be defined in terms of its power series.

Since the computation of this exponential for a tensor of arbitrary rank becomes rather complicated, we will illustrate how the procedure works for scalars and vectors. For a scalar, we find

$$[\Box, \nabla_\alpha] X = -\frac{R}{d}\nabla_\alpha X , \tag{74}$$

such that for $\mathbf{C}$, we obtain

$$C_\alpha{}^\beta = -\frac{R}{d}\delta_\alpha{}^\beta . \tag{75}$$

Consequently, inserting this into (73), we have

$$f(\Box)\nabla_\alpha X = \nabla_\alpha f\left(\Box - \frac{R}{d}\right)X . \tag{76}$$

The same procedure can be applied for vectors. First, we find for **C**

$$C_{\alpha\mu}{}^{\beta\nu} = -\frac{R}{d}\delta_\alpha{}^\beta + \frac{2}{d-1}\frac{R}{d}g_{\alpha\mu}g^{\beta\nu} - \frac{2}{d-1}\frac{R}{d}\delta_\alpha{}^\beta\delta_\mu{}^\nu, \tag{77}$$

which gives for (73)

$$\begin{aligned}
f(\Box)\nabla_\alpha X_\mu &= \nabla_{(\alpha}f\left(\Box - \frac{d+1}{d-1}\frac{R}{d}\right)X_{\mu)} + \frac{1}{d}g_{\alpha\mu}\nabla^\beta\left[f\left(\Box + \frac{R}{d}\right) - f\left(\Box - \frac{d+1}{d-1}\frac{R}{d}\right)\right]X_\beta \\
&\quad + \nabla_{[\alpha}f\left(\Box - \frac{d-3}{d-1}\frac{R}{d}\right)X_{\mu]}.
\end{aligned} \tag{78}$$

The formula for the rank-two tensor can be found in the same fashion. However, since this is very lengthy, we will not display it here.

## C Differential equations for $(\Box + z)^{-1}T_{\mu\nu}$

In this appendix, we collect several details regarding the calculation of the propagator in the 2-derivative sector.

We begin with a complete list of the differential equations for the propagator functions occurring in the ansatz (45). These differential equations read

$$\begin{aligned}
\eta^2 G_{00}'' + (d+4)\eta G_{00}' + \left(q^2\eta^2 + \zeta + 4\right)G_{00} &= \frac{1}{H^2\eta^2} - 2(d-1)G_\delta - 4\vec{p}_1\cdot\vec{p}_2 G_{(12)} \\
&\quad - 2p_1^2 G_{11} - 2p_2^2 G_{22} + 4i\eta(p_1^2 - \vec{p}_1\cdot\vec{p}_2)G_1 - 4i\eta(p_2^2 - \vec{p}_1\cdot\vec{p}_2)G_2, \\
\eta^2 G_1'' + (d+4)\eta G_1' + \left(q^2\eta^2 + \zeta + d\right)G_1 &= 2i\eta\big(G_{00} - G_\delta \\
&\quad - (p_1^2 - \vec{p}_1\cdot\vec{p}_2)G_{11} + (p_2^2 - \vec{p}_1\cdot\vec{p}_2)G_{(12)}\big), \\
\eta^2 G_2'' + (d+4)\eta G_2' + \left(q^2\eta^2 + \zeta + d\right)G_2 &= 2i\eta\big(-G_{00} + G_\delta \\
&\quad - (p_1^2 - \vec{p}_1\cdot\vec{p}_2)G_{(12)} + (p_2^2 - \vec{p}_1\cdot\vec{p}_2)G_{22}\big), \\
\eta^2 G_{11}'' + (d+4)\eta G_{11}' + \left(q^2\eta^2 + \zeta + 2d\right)G_{11} &= -4i\eta G_1, \\
\eta^2 G_{22}'' + (d+4)\eta G_{22}' + \left(q^2\eta^2 + \zeta + 2d\right)G_{22} &= 4i\eta G_2, \\
\eta^2 G_{(12)}'' + (d+4)\eta G_{(12)}' + \left(q^2\eta^2 + \zeta + 2d\right)G_{(12)} &= 2i\eta\left(G_1 - G_2\right), \\
\eta^2 G_\delta'' + (d+4)\eta G_\delta' + \left(q^2\eta^2 + \zeta + 2d\right)G_\delta &= -2G_{00}.
\end{aligned} \tag{79}$$

We now perform linear transformations such that the system of differential equations becomes partially decoupled. Defining $\mathfrak{q} = -q\eta$, we choose coefficients $a_{ij}$ such that $g_i(\mathfrak{q}) = \sum_j a_{ij}G_j(\eta)$, and the differential equations are cast in the following form:

$$\begin{aligned}
\mathfrak{q}^2 g_1'' + (d+4)\mathfrak{q}g_1' + (\mathfrak{q}^2 + \zeta + d + 2)g_1 &= \frac{1}{\mathfrak{q}^2} + dg_2, \\
\mathfrak{q}^2 g_2'' + (d+4)\mathfrak{q}g_2' + (\mathfrak{q}^2 + \zeta + d)g_2 &= \frac{1}{\mathfrak{q}^2} + (d-2)g_1 + 4i\mathfrak{q}g_3, \\
\mathfrak{q}^2 g_3'' + (d+4)\mathfrak{q}g_3' + (\mathfrak{q}^2 + \zeta + d)g_3 &= 4i\mathfrak{q}g_2, \\
\mathfrak{q}^2 g_4'' + (d+4)\mathfrak{q}g_4' + (\mathfrak{q}^2 + \zeta + 2d)g_4 &= -2i\mathfrak{q}g_5, \\
\mathfrak{q}^2 g_5'' + (d+4)\mathfrak{q}g_5' + (\mathfrak{q}^2 + \zeta + d)g_5 &= -2i\mathfrak{q}g_4, \\
\mathfrak{q}^2 g_6'' + (d+4)\mathfrak{q}g_6' + \left(\mathfrak{q}^2 + \zeta + 2(d+1)\right)g_6 &= \frac{1}{\mathfrak{q}^2}, \\
\mathfrak{q}^2 g_7'' + (d+4)\mathfrak{q}g_7' + (\mathfrak{q}^2 + \zeta + 2d)g_7 &= 0.
\end{aligned} \tag{80}$$

Note that there are now four decoupled systems: $\{g_1, g_2, g_3\}$, $\{g_4, g_5\}$, and the separate equations for $g_6$ and $g_7$. For brevity, we will refrain from displaying the coefficients $a_{ij}$ here; these are given in the attached notebook.

## C.1  Details regarding solving for $g_{1,2,3}$

We will now focus on the subsystem of equations in (80) for $g_{1,2,3}$, since these contribute to the function $G_{00}$ in the leading-order expansion in $\mu_\chi$. This system is solved by inserting the ansatz

$$g_{i,\nu}(\mathfrak{q}) = \mathfrak{q}^\nu \sum_{k \geq 0} a_{i,k} \mathfrak{q}^n. \tag{81}$$

Solving differential equations in this way is also known as the Frobenius method [123–127]. Inserting the ansatz in the differential equations, and collecting powers of $\mathfrak{q}$ allows to read off the following coupled recursion relations for the coefficients $a_{i,k}$:

$$
\begin{aligned}
0 &= a_{1,k-2} + \big((k+\nu)^2 + (d+3)(k+\nu) + \zeta + d + 2\big) a_{1,k} - d a_{2,k}, \\
0 &= a_{2,k-2} + \big((k+\nu)^2 + (d+3)(k+\nu) + \zeta + d\big) a_{2,k} - (d-2) a_{1,k} - 4\mathrm{i} a_{3,k-1}; \\
0 &= a_{3,k-2} + \big((k+\nu)^2 + (d+3)(k+\nu) + \zeta + d\big) a_{3,k} - 4\mathrm{i} a_{2,k-1}.
\end{aligned}
\tag{82}
$$

These equations can be decoupled by substituting one equation into the other. Eventually, this yields three equations that each depend on $a_{i,k}$, $a_{i,k-2}$, $a_{i,k-4}$, $a_{i,k-6}$. We solve these equations using Mathematica's `RSolve` method. The resulting power series can be resummed in terms of generalised hypergeometric functions:

$$g_{i,\nu}(\mathfrak{q}) = \mathfrak{q}^\nu \sum_{j=1,2,3} c_{i,\nu,j} \sum_{\ell=1,2,3} \alpha_{i,j,\ell}\, {}_1\tilde{F}_2\left(\beta_{i,j,\ell}; \gamma_{i,j,\ell}, \delta_{i,j,\ell}; -\frac{\mathfrak{q}^2}{4}\right), \tag{83}$$

where $\beta_{i,j,\ell} \in \{1,2\}$, while $\alpha_{i,j,\ell}$, $\gamma_{i,j,\ell}$ and $\delta_{i,j,\ell}$ depend on $\nu$ and $\zeta$. Their values can be found in the accompanying notebook. The coefficients $c_{i,\nu,j}$ are free parameters, that remain to be fixed.

The system of three second-order linear differential equations has a six-dimensional solution space. We will consider only the inhomogeneous solution; adding a homogeneous solution will generically lead to non-analyticities. The inhomogeneous solution is given by

$$g_1(\mathfrak{q}) = g_{1,-2}(\mathfrak{q}), \qquad g_2(\mathfrak{q}) = g_{2,-2}(\mathfrak{q}), \qquad g_3(\mathfrak{q}) = g_{3,-3}(\mathfrak{q}) + g_{3,-1}(\mathfrak{q}). \tag{84}$$

At this stage, a remark on the parameter values $\nu \in \{-3, -2\}$, $\zeta = 2$ is in order. The functions (83) are singular at these values. This is linked to the massless nature of the graviton. The divergence is removed by carefully choosing part of the parameters $c_{i,\nu,j}$; the other parameters are fixed by inserting the functions (84) into the differential equation. For $\zeta \neq 2$, the inhomogeneous solution to the differential equation is given by $g_3 = g_{3,-1}$, while $g_1$ and $g_2$ in (84) remain unaltered. In this case, all of the coefficients $c_{i,\nu,j}$ can be found by inserting the $g_i$ into the differential equation.

Returning to the case $\zeta = 2$, we have now found the solution to the differential equation. These are given in terms of generalised hypergeometric functions, as well as derivatives thereof with respect to their parameters. Since the total expressions are rather complicated, we will refrain from listing them here, and refer to the attached notebook for their explicit expressions.

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
