# Peer review of "De Sitter scattering amplitudes in the Born approximation"

_SciPost Physics, doi:SciPost Phys. 13, 106 (2022)_

## Round 2 · Referee Report · Anonymous (Referee 1) · 2022-4-4

Strengths

1) The topic of tree-level scattering amplitudes in de Sitter spacetime is of direct physical relevance to cosmology. The underlying connection to form factors in a quantum effective action is of great importance to build connections from quantum-gravity scenarios to observable physics.

2) The methodology, assumptions, and aims of the work are very clearly presented. The calculations are presented in great detail and thanks to attached supplementary files easy to reproduce.

Weaknesses

1) In some cases (see change request below for the most important places), the authors could elaborate more on the physical implications / interpretations of their results.

2) The paper would benefit from a more extensive comparison to existing calculations (many of which the authors cite in the introduction) of tree-level scattering amplitudes. In particular, a comparison of the results (not just in flat Minkowski spacetime) would be of value.

3) The paper would appear stronger if a technical explanation (in a sense of a mathematical explanation at the level of Fourier methods) for the vanishing of the potential beyond unit dimensionless proper distance can be added.

Report

The authors' work provides an interesting pathway to connect cosmological observations to tree-level scattering amplitudes in curved spacetime. An extension to general form factors has the potential to link quantum gravity approaches to cosmology and therefore observation.

The methods and calculations in this paper are very clearly presented and can all be readily reproduced.

The authors' results would gain impact if the contextualization with existing literature and the physical conclusions would be strengthened. In particular, the striking result of an oscillating tree-level amplitude and a potential with discontinuous derivative call for a better understanding.

With these improvement (see requested changes below), I recommend the paper for publication in SciPost.

Requested changes

There are three main points which deserve improvement:

1) The authors provide a nice discussion of previous work on scattering amplitudes in de Sitter spacetime. However, it remains unclear to the reader how their results compare to these previous works. Do other gauge-invariant results for the graviton-mediated tree-level amplitude in de Sitter spacetime exist? How do the central results in sections 3 and 4 compare to previous results? (Or why is such a comparison not meaningful?) Such a comparison should also be addressed in the conclusions. 2) The authors choice of units should be clarified. When $\mu$ is first introduced in the introduction H is not given in natural units and $\mu$ is therefore not dimensionless. Further, the units of the amplitude functional and the scattering potential should be clarified. 3) Previous applications of the heavy-mass limit (if they exist) should be carefully reviewed and discussed. In particular, with view on whether this approximation is expected to hold for all momenta/distances in the scattering amplitude/potential. In this context, the "very massive scalars" in the abstract would better be stated in comparison to a second mass scale. 4) Around Eq. (50), the authors discuss the choice of $\zeta=2$. It remains unclear whether this choice has a physical motivation. What follows from other choices? In particular, because this choice seems to be used in order to obtain the main final results: (i) the amplitude behaviour in Fig 2 and (ii) the scattering potential in Fig. 3. 5) Can the authors explain the vanishing of the scattering potential at $\frak{r}>1$ at the mathematical level in terms of Fourier theory? Such an explanation would make the result more convincing. 6) In the conclusions, the authors use the terminology "operator methods" which is not used elsewhere in the paper. A more complete explanation and use of the same terminology as in the main text will help the reader to understand what is at the core of their novel methods. 7) In many places (including the abstract, conclusions, and discussions of physical implications), the authors use the term "scattering amplitude" when, I think, they should clarify that they calculate the "tree-level scattering amplitude". Is it possible that some of the physical implications are altered when including terms beyond tree-level?

Here are a couple of minor remarks: * "the the Ads/CFT correspondence" (double the) * App A is not mentioned in the outline at the end of Sec 1 * citations for the background field method and de Donder type gauge fixing would help the reader * a clarification of derivative operators with and without hat/bar would be useful, in particular, in the calculation from (15) onwards * below (10), can the authors exclude other stationary points? and if not, what would be their significance? * below (17), what is meant by "trivial index structure"? * in (23) and onward the authors may want to clarify the meaning of an overbar (presumably complex conjugation) * if the authors state that (26) is well-known, they may want to add a citation * the statement below (34) about the Bunch-Davies vacuum may also be deserving of a citation * "with ... at hand" in the beginning of Sec 4.2: a reference to the respective equations would be helpful

  • validity: high
  • significance: good
  • originality: high
  • clarity: high
  • formatting: excellent
  • grammar: excellent

Author:  Chris Ripken  on 2022-06-07  [id 2564]

(in reply to Report 1 on 2022-04-04)
Category:
answer to question
correction
pointer to related literature

We would like to thank the referee for their detailed comments and remarks. Below, we have addressed their comments point-by-point.

1) The referee recommends elaborating more on previous results, and to make a comparison to our work. In the case of the amplitude functional, this is not straightforward, since our use of a novel technique to compute this object. We have commented on previous computations in section 3.5. In relation to the scattering potential, we have added a paragraph in section 4.5 to address previous works.

2) The referee remarks that our choice of units in the introduction and subsequent sections is not clear. We have added a paragraph in the introduction regarding units in de Sitter spacetime, and added a comment on the units of the scattering amplitude and potential in sections 4.4 and 4.5.

3) The referee advises to add a discussion of previous applications of the heavy-mass limit. After the submission of this work, we have investigated the connection of this expansion to the adiabatic approximation. We have added a short discussion of the adiabatic approximation in sect. 4.1. A detailed discussion of the adiabatic approximation will appear in a subsequent work.

4) The referee requests a clarification of the choice $\zeta =2$ in eq. (50). As we explain, the differential operator that appears in eq. (18) is of the form $(\square + \zeta H^2)^{-1}$, with $\zeta=2$. Here the value of $\zeta$ denotes the ``mass'' of the graviton. The graviton mass is fixed by the action described in section 3.1.

In order to obtain scattering amplitudes with different values of $\zeta$, one has to add additional terms to the action. In a future paper, we will discuss quadratic gravity, which includes an $R^2$ and a $\text{Weyl}^2$ term in the action. The associated couplings give rise to operators $(\square + \zeta H^2)^{-1}$ where $\zeta$ can be chosen freely by tuning the couplings.

Furthermore, as we remark below eq. (50) and elaborate on in appendix C, the choice $\zeta=2$ is special because the solutions to the power series ansatz in general contains singularities. These are fixed by carefully choosing the coefficients of the power series ansatz. We suspect that these singularities are linked to the masslessness of the graviton, which should mean that the graviton is in a different representation of the de Sitter group (comparable to a massless particle being in a different representation of the Lorentz group in Minkowski spacetime than a massive particle). A discussion of the representation theory of the de Sitter group is beyond the scope of this work, and is left to future work.

5) The referee remarks that an analytic derivation of the vanishing of the potential for $\mathfrak{r}>1$ would make the result more convincing. The vanishing potential is the result of evaluating eq. (58), which is a rather complicated Mellin transform. We have added a reference to a work where this Mellin transform is computed using analytic methods.

6) The referee remarks that the nomenclature "operator methods" is introduced only in the conclusion. We have added a clarification of this terminology in the introduction.

7) The referee points out that the term "scattering amplitude" has been used where a phrasing "tree-level scattering amplitude" would be more apt. We have carefully reconsidered the wording throughout the paper and made adjustments where this is appropriate.

The referee asks what the implications could be beyond tree-level. At this stage, we have not made any investigations into this direction. Including loop effects would give the quantum corrections to the scattering amplitude. We expect that for short distances, these are consistent with Effective Field Theory, giving logarithmic divergences. At large distances, however, the modifications are at this stage unknown.

Furthermore, the referee asks some clarifications: - The referee asks whether it is possible to exclude other stationary points than vanishing scalar fields. A straightforward generalisation here would be to assign a vacuum expectation value to the scalar field, setting $\phi = c$. In our setup, this is not possible since this is not an solution to the equation of motion. It would be interesting to introduce a scalar self-coupling and study the impact of a vev (e.g. in the context of symmetry breaking). - The referee recommends to add a citation regarding the statement that the Bunch-Davies vacuum is the unique vacuum such that the function $\mathcal{E}$ has a simple pole in the mass, similar to the Minkowski vacuum. To the best of our knowledge, this particular statement has not been made elsewhere. Usually, the Bunch-Davies vacuum is defined using asymptotic boundary conditions, or adiabaticity conditions, which are slightly different in nature.

Apart from these remarks, the referee made a few minor remarks, which we have adjusted in the paper.

---

## Round 2 · Referee Report · Anonymous (Referee 2) · 2022-4-12

Report

In this well-written work the authors compute the potential of a graviton-mediated scattering process involving two very massive scalars in dS geometry. The results are interesting and deserve publication, I only have one remark.

The main result of the paper is Eq.(20), whose derivation is described in Appendix B. The mathematica package implements the covariant approach by DeWitt. This is based on a local expansion based on the BCH formula, and the question is if the conclusions of the paper can be trusted also in the deep IR, near the Hubble scale.

I would like the authors in the new version of the paper to discuss the limitations of their approach in connection with the claim of a repulsive force at sub-Hubble distance. I do no need to see the paper again.
  • validity: -
  • significance: -
  • originality: -
  • clarity: -
  • formatting: -
  • grammar: -

Author:  Chris Ripken  on 2022-06-07  [id 2563]

(in reply to Report 2 on 2022-04-12)
Category:
answer to question

We thank the referee for helpful comments on our manuscript. In this reply, we will address their points raised above.

The referee remarks that our expansion is local. This is indeed the case; the expansion in $\mu = m/H \gg 1$ is at least asymptotic. At this stage, we have not investigated the actual convergence of this expansion. Hence, the results at large distances should be taken with some care.

The interpretation of the modified potential as a repulsive force comes from the expansion at $\mathfrak{r}=0$. Hence, this describes the short-distance behaviour. If the expansion around large masses is reliable, this observation should be robust.

We have added a paragraph to the manuscript to address these comments in the conclusion.

---

## Round 3 · Author Response

We would like to thank the referees for giving constructive comments on our work. We have addressed the comments in the respective replies to the referees.

---

## Round 3 · List of Changes

We refer here to the replies to the referees for the main changes made in the manuscript.
Apart from this, we fixed a number of typos.

The main changes are:
- The terminology on tree-level scattering amplitudes and operator methods was made consistent.
- We have extended our discussion of our results with respect to existing literature.

---

## Editorial Decision

published